# ADHD Disease Detection Based on Short- and Long-Term Brain Function Encoding and Memory Graph Network

**Dongxun Jiang**[1]  **Borui Jia**[1]  **Yuxuan Wang**[1]  **Dongdong Zhang**[2]

## Abstract

Graph-based attention deficit hyperactivity disorder (ADHD) detection methods have been extensively studied, but comparatively less attention has been paid to short-term brain functional reorganization. In this paper, we propose an ADHD disease detection model based on short- and long-term brain function encoding and memory graph network. We first exploit a novel brain map sequence construction method based on short-term windows to extract short-term brain function features. Then, we design a short-term state and temporal dependency encoder to characterize short-term sequence patterns of brain function. Furthermore, a brain function memory is introduced to capture the association of brain activity patterns and historical sequence patterns. Concurrently, GNN-based long-term brain function feature extraction network is used to extract brain structure features, which are fused with short-term features for ADHD detection. Experimental validation on the publicly available neuroimaging datasets ADHD-200 and OpenNeuro-ds002424 demonstrates the superior performance of our model in brain disorder detection.

## 1. Introduction

Attention-Deficit/Hyperactivity Disorder (ADHD) is one of the most common neurodevelopmental disorders in childhood and adolescence, with a global prevalence of approximately 5–7% (Polanczyk et al., 2014). Functional connectomics models the brain as a graph, where nodes represent brain regions and edges represent the strength of functional connectivity between regions, providing a powerful mathe-

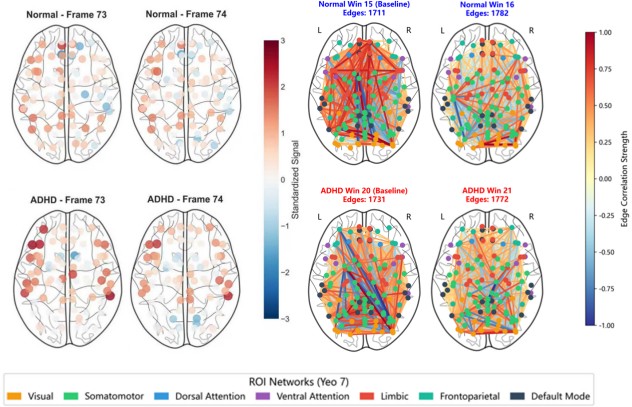

*Figure 1.* Comparison of our short-term brain graph sequences and adjacent frame time series in EBIGNN. EBIGNN uses changes in brain region nodes for disease diagnosis. We employ brain map sequences based on short-term windows to characterize short-term neural activity changes in the brain. For a detailed comparison, please refer to Figure 18 and 19 in Appendix B.

matical framework for systematically analyzing brain network organization (Bullmore & Sporns, 2009).

Graph analysis based on static functional connectivity has yielded a series of significant findings in ADHD research. Graph theory analysis reveals that the whole-brain networks of ADHD patients exhibit abnormal topological characteristics (Wang et al., 2009; Cocchi et al., 2012). Network hierarchy analysis revealed weakened antagonistic relationships between the default mode network (DMN) and task-related networks (TPNs) (Castellanos et al., 2008; Fair et al., 2010). Machine learning approaches utilizing whole-brain or network-specific connectivity strength as features have demonstrated promising performance in ADHD classification tasks (Colby et al., 2012). While static connectivity methods make significant contributions to revealing network-level abnormalities in ADHD, they suffer from two fundamental limitations: the absence of temporal information and coarse spatial resolution.

In recent years, Graph Neural Networks (GNNs) have demonstrated powerful potential for processing brain connectivity data. By aggregating and propagating information across graph structures, GNNs learn embedded repre-

[1]Guohao College, Tongji University, Shanghai, China [2]School of Computer Science and Technology, Tongji University, Shanghai, China. Correspondence to: Dongdong Zhang <ddzhang@tongji.edu.cn>.

*Proceedings of the 43$^{rd}$ International Conference on Machine Learning*, Seoul, South Korea. PMLR 306, 2026. Copyright 2026 by the author(s).

sentations of nodes (brain regions) while considering the features and connectivity of neighboring nodes (Kipf & Welling, 2017). Architectures such as Graph Convolutional Networks (GCNs) and Graph Attention Networks (GATs) have been successfully applied to disease classification using static functional connectivity, achieving superior performance compared to traditional machine learning methods in identifying conditions like ADHD, autism, and Alzheimer's disease (Ktena et al., 2018; Wen et al., 2022). The advantage of these methods lies in their ability to learn end-to-end mappings from raw connectivity matrices to diagnostic labels, automatically extracting discriminative features without requiring manual design of graph metrics.

To combine the representational power of GNNs with the temporal information of dynamic connectivity, researchers have begun exploring graph sequence modeling approaches. These approaches treat dynamic connection sequences as time-varying graphs, employing recurrent neural networks (RNNs) or variants to capture temporal dependencies while using graph convolutions to extract spatial features at each time step (Yan et al., 2018). For example, Spatio-Temporal Graph Convolutional Networks (ST-GCN) apply graph convolutions at each time step to extract spatial features, followed by 1D convolutions or RNNs along the temporal axis to capture temporal evolution (Gadgil et al., 2020). Effective Brain Inference Graph Neural Network(EBIGNN) adaptively aggregates dynamic graph structures across different time points through a temporal attention mechanism, explicitly encoding causal temporal information between adjacent frames into downstream graph learning tasks (Mamoon et al., 2025). These methods have shown promising results in classification tasks for disorders, suggesting that temporal information contains disease-related discriminative features.

Although current research has characterized dynamic changes in brain regions between adjacent frames across the temporal dimension, there remains insufficient focus on extracting features of short-term functional reorganization and transient activity. Additionally, there is a lack of effective integration of local and global spatiotemporal information to reveal disease-related discriminative features.

To address the challenges, we propose a novel ADHD disease detection model based on short- and long-term brain function encoding and memory graph network (SLT-BFGN). We exploit a novel brain map sequence method based on short-term windows to extract short-term brain function features. Figure 1 presents a comparison between the adjacent frame time series method and ours. Short-term state and temporal dependency encoder is designed to characterize the dynamic sequential patterns of brain function. A brain function memory is introduced to associate transient brain activity patterns and historical sequence patterns. GNN-based long-term brain function feature extraction network

is employed to derive structural brain attributes. These features are fused with short-term features to enhance ADHD detection. Our model demonstrates outstanding detection performance on the ADHD-200 and OpenNeuro-ds002424 datasets.

The main contributions of this work are summarized as follows:

- We propose ADHD disease detection model based on short- and long-term brain function encoding and memory graph network (SLT-BFGN), in which a novel brain map sequence method based on short-term windows is exploited to extract short-term brain function features.

- We design a short-term state and temporal dependency encoder to characterize the dynamic sequential patterns of brain function, as well as a brain function memory to associate brain activity patterns and historical sequence patterns.

- We introduce GNN-based long-term brain function feature extraction network to derive structural brain features. These features are fused to enhance the performance of ADHD detection with short-term features to enhance ADHD detection.

## 2. Related Work

### 2.1. Static Global Graph-Based Methods in Brain Disease Detection

Diagnosing brain diseases is inherently challenging due to the brain's complex, dynamic, and highly interconnected structure. Traditional methods that use static global graphs have difficulty capturing the nuanced, nonlinear relationships present in brain connectivity data, especially when integrating multimodal sources such as fMRI, MRI, and clinical/genetic information. GroupBNA (Peng et al., 2024) incorporates a group-adaptive network enhancement strategy into the graph network to enhance collective perception capabilities in the diagnosis of brain disorders. BrainOOD (Xu et al., 2025) is introduced as a novel framework tailored for brain networks that enhances GNNs' out-of-distribution (OOD) generalization and interpretability. To further analyze and enhance the structural characteristics of brain disease detection networks, GNNMA (Si et al., 2025) introduces a modular attention mechanism into GATs and Graph-based Networks (BNTs) via Singular Value Decomposition (SVD) to reduce the dimensionality of high-dimensional graph data while preserving critical information, thereby more effectively capturing modular features within graphs. However, current methods for extracting dynamic and static features from static global brain maps are confined to the entire MRI sequence, lacking the ability to extract neural activity within short-term windows.

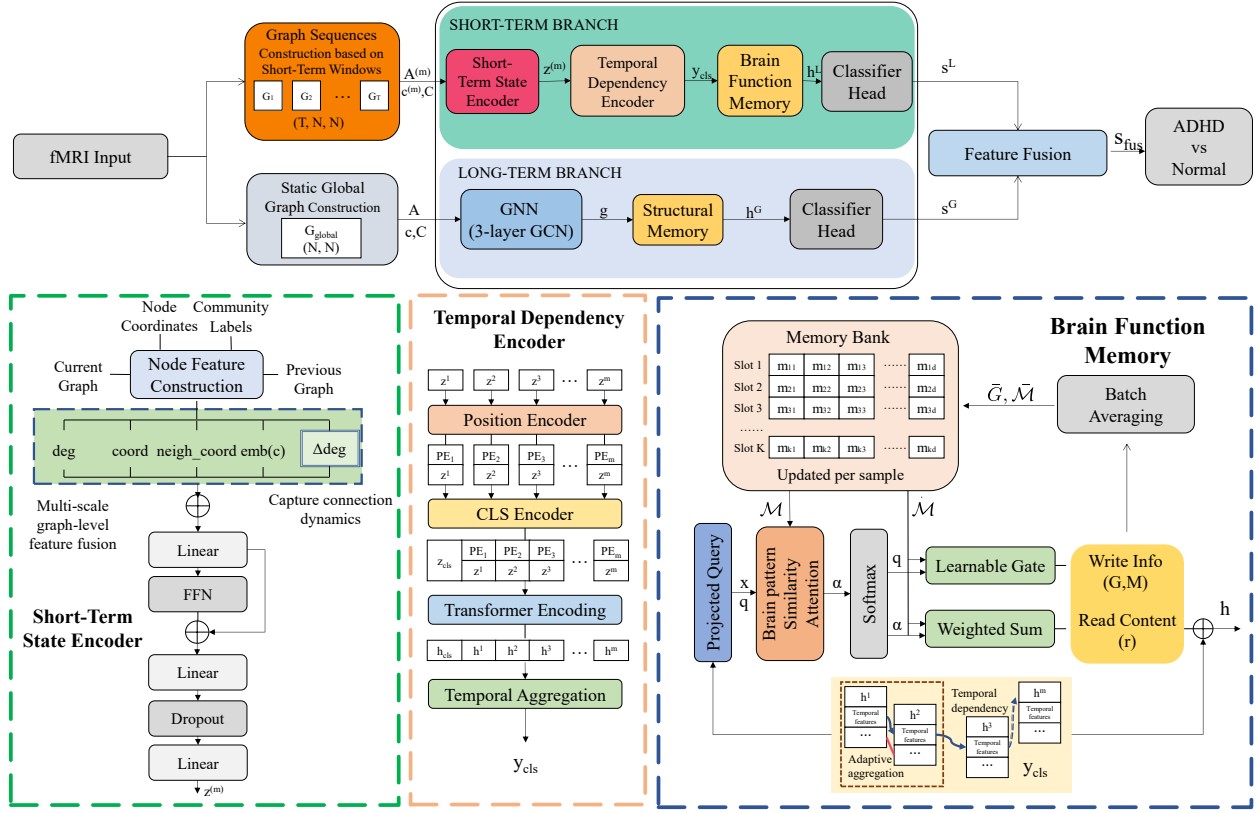

*Figure 2.* Overview of our model.

## 2.2. Graph Time Series-Based Methods in Brain Disease Detection

Graph time series methods model dynamic changes in brain connectivity, which is crucial for tracking disease-related neural dynamics. A DyGAE-based framework (Noman et al., 2022) leverages time-varying topological structures of dynamic brain networks. EBIGNN (Mamoon et al., 2025) infers dynamic effective connectivity between adjacent MRI frames to better characterize brain networks. ET_MGNN (Lang et al., 2025) uses an overlapping sliding window and combines RNNs with Transformers to capture temporal dependencies, forming a unified brain network representation from dynamic functional and structural connectivity. Although graph time series methods have made progress in brain disease detection, current approaches focus on the direct relationships between adjacent MRI frames, making it difficult to characterize changes within local time windows and the functional connectivity between neural regions.

## 3. Methods

### 3.1. Motivation and Overview

The neural mechanisms underlying ADHD are complex and dynamically variable. Despite advances in dynamic func-

tional connectivity research, existing methods remain inadequate in capturing brain functional reorganization within short-term windows. Therefore, we propose a novel ADHD disease detection model based on short- and long-term brain function encoding and memory graph network (SLT-BFGN) to more comprehensively reveal the dynamic characteristics of brain networks in ADHD. Figure 2 describes the overview of our model.

Neurodynamic analysis across subjects from independent sites reveals that brain functional connectivity evolution approximately follows a first-order Markov process, where lag-1 autocorrelation is $\rho_1 = 0.336$ while lag-2 decays to near zero (Markov ratio $R = 4.76$), which is further validated in Section 4.4 and detailed in Appendix A. This formally motivates the short-term window design, as single-frame methods discard this sequential structure entirely, while long-horizon methods model dependencies that provably do not exist in the data.

Based on neurodynamic analysis, a novel brain map sequence construction method based on short-term windows is first proposed to describe short-term brain neural activity. Then, a short-term state and temporal dependency encoder is proposed to characterize short-term sequential brain function patterns. A brain function memory module is designed

to capture the associations between brain activity patterns and historical sequential patterns. Concurrently, GNN-based long-term brain function feature extraction network is introduced to derive structural brain features. These features are then integrated with short-term functional features for the detection of ADHD.

## 3.2. Construction of Brain Map Sequences Based on Short-Term Windows

To describe short-term functional reorganization and transient activity in the brain, we construct short-term functional window sequences of brain maps as shown in Figure 3.

Raw fMRI data are parcellated into $N$ ROIs using the AAL atlas, followed by standard preprocessing. We divide the preprocessed ROI time-series matrix $X \in \mathbb{R}^{T \times N}$ into $M = \lfloor T/w \rfloor$ non-overlapping windows of length $w$. The $m$-th window is $X^{(m)} = X[s_m : s_m + w, :]$ with $s_m = (m-1) \cdot w$. For each window $X^{(m)}$, we estimate a precision matrix $\Theta^{(m)}$ using robust covariance estimation techniques with appropriate regularization to ensure numerical stability.

Then we convert $\Theta^{(m)}$ into a window-level partial correlation matrix $P^{(m)}$ (diagonal set to 0) as

$$P_{ij}^{(m)} = -\frac{\Theta_{ij}^{(m)}}{\sqrt{\left|\Theta_{ii}^{(m)}\right|\left|\Theta_{jj}^{(m)}\right| + \varepsilon}}, \quad i \neq j, \qquad (1)$$

where $\varepsilon$ is a small constant for numerical stability. This directly matches our edge weight definition: stronger conditional dependence implies larger $|P_{ij}^{(m)}|$.

To make all windows comparable under the same sparsification criterion, we pool the absolute upper-triangular values across all windows and compute a global percentile threshold $\tau$. Then each window graph is obtained by hard-thresholding:

$$A_{ij}^{(m)} = \begin{cases} P_{ij}^{(m)}, & i \neq j \text{ and } |P_{ij}^{(m)}| \geq \tau, \\ 0, & \text{otherwise.} \end{cases} \qquad (2)$$

We then enforce symmetry by $A^{(m)} \leftarrow (A^{(m)} + (A^{(m)})^T)/2$. Here, $p$ controls the density of each window graph, and using a global $\tau$ ensures the sparsity level is aligned across short-term windows, which is crucial for learning temporal brain-network dynamics.

For each window graph $A^{(m)}$, modularity maximization is performed to extract community labels $c^{(m)} \in \mathbb{Z}^N$. The final output includes the time-indexed graph sequence $\{A^{(m)}\}_{m=1}^M$, the community sequence $\{c^{(m)}\}_{m=1}^M$, and window start indices.

Regarding the above construction process, we conduct temporal dependency analysis (Appendix A) and dynamic functional connectivity analysis (Appendix B), demonstrating

significant temporal dependency in brain graph sequences and the effectiveness of short-term window-based sequences from the perspective of neurodynamics.

## 3.3. Short-Term State and Temporal Dependency Encoder

To characterize short-term sequence patterns of brain function, a short-term state and temporal dependency encoder is designed, consisting of a short-term state encoder and a Transformer-based temporal dependency encoder.

### 3.3.1. SHORT-TERM STATE ENCODER

For each window $m$, we construct node-level features by combining degree statistics, spatial information, and topological clustering.

**Degree Dynamics and Neighborhood Geometry.** For node $i$ in window $m$, we compute the absolute degree and its short-term change:

$$\begin{aligned} deg_i^{(m)} &= \sum_{j=1}^N \left|A_{ij}^{(m)}\right|, \\ \Delta deg_i^{(m)} &= deg_i^{(m)} - deg_i^{(m-1)}, \end{aligned} \qquad (3)$$

and define the neighborhood geometric center as

$$neigh\_coord_i^{(m)} = \sum_{j=1}^N \frac{A_{ij}^{(m)}}{\max(deg_i^{(m)}, 1)} \, coord_j, \quad (4)$$

where $coord_j \in \mathbb{R}^3$ is the 3D coordinate of ROI $j$. Intuitively, $deg_i^{(m)}$ captures current connectivity strength, $\Delta deg_i^{(m)}$ captures short-term reorganization, and $neigh\_coord_i^{(m)}$ summarizes the spatial center of $i$'s connected neighbors weighted by partial correlations.

**Feature Aggregation.** The raw node feature vector is $f_i^{(m)} = [deg_i^{(m)}, \ coord_i, \ neigh\_coord_i^{(m)}, \ emb(c_i^{(m)}), \ \Delta deg_i^{(m)}]$, where $emb(c_i^{(m)})$ denotes the learnable community embedding. These features are transformed through layer normalization, linear projection, and residual feed-forward networks. Average pooling over all nodes yields the window-level embedding $z^{(m)} = \frac{1}{N} \sum_{i=1}^N x_i^{(m)}$, where $x_i^{(m)}$ is the encoded node representation. This $z^{(m)}$ serves as a compact representation of the brain state in window $m$.

**Community Anomaly Statistics.** Additionally, we compute community-based anomaly scores $s^{(m)}$ to capture rare community patterns. Community labels with lower frequency in the training set receive higher anomaly scores:

$$s^{(m)} = -\frac{1}{|\mathcal{V}^{(m)}|} \sum_{i \in \mathcal{V}^{(m)}} \log \pi(c_i^{(m)}), \qquad (5)$$

where $\pi(\cdot)$ is the empirical community frequency and $\mathcal{V}^{(m)} = \{i \mid c_i^{(m)} \neq -1\}$. These window-level statis-

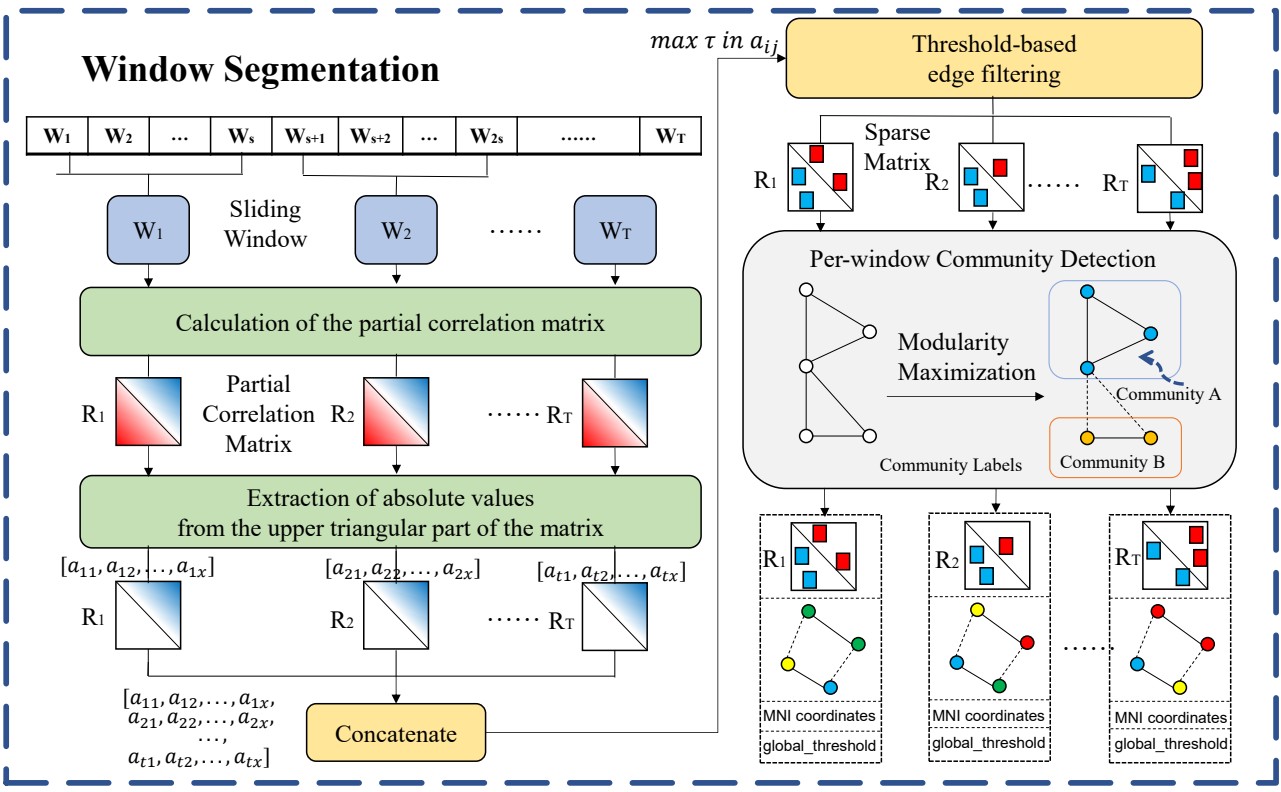

*Figure 3.* Construction of brain map sequences based on short-term windows.

tics are aggregated across the sequence and incorporated into the short-term branch features.

### 3.3.2. TEMPORAL DEPENDENCY ENCODER

To capture temporal dependencies across the graph sequence, we employ a Transformer-based encoder architecture with explicit positional encoding.

**Positional Encoder.** Given $M$ window-level embeddings $\{z^{(1)}, z^{(2)}, \ldots, z^{(M)}\}$ (each $z^{(m)} \in \mathbb{R}^d$) from the short-term state encoder, we first inject positional information to explicitly encode temporal order. We define the positional encoding $\text{PE}^{(m)} \in \mathbb{R}^d$ using standard sinusoidal functions, where $m \in \{1, 2, \ldots, M\}$ is the window index. The positionally-encoded window representation is then obtained by element-wise addition: $z'^{(m)} = z^{(m)} + \text{PE}^{(m)}$. This step assigns unique positional information to the originally order-agnostic window features, enabling the model to distinguish temporal order.

**Classification Token Construction.** To aggregate sequence-level information, we introduce a learnable classification token $z_{\text{cls}} \in \mathbb{R}^d$ that is randomly initialized and optimized during training. This token is prepended to the positionally-encoded window sequence: $S = [z_{\text{cls}}; z'^{(1)}; z'^{(2)}; \ldots; z'^{(M)}] \in \mathbb{R}^{(M+1) \times d}$. For variable-

length sequences, we apply a padding mask to prevent attention to padded positions.

**Transformer Encoding.** The input sequence $S$ is processed through $L$ stacked Transformer encoder layers. Each layer $\ell \in \{1, 2, \ldots, L\}$ consists of multi-head self-attention and position-wise feedforward networks with residual connections. The multi-head self-attention mechanism enables each position to attend to all other positions, thereby modeling temporal dependencies between windows.

**Temporal Aggregation.** After $L$ layers of encoding, we extract the output at the classification token position as the final temporal-encoded representation: $y_{\text{cls}} = S^{(L)}[0,:] \in \mathbb{R}^d$. Through the attention mechanism, $y_{\text{cls}}$ aggregates information from all $M$ positionally-encoded windows, encoding both instantaneous brain states and their temporal evolution patterns.

### 3.4. Brain Function Memory

To capture the association of transient brain activity features with historical sequence features, we introduce a learnable memory module in the short-term branch. The memory maintains a learnable memory matrix $\mathcal{M} \in \mathbb{R}^{K \times d_m}$ with $K$ memory slots, each storing a $d_m$-dimensional pattern

vector. This external memory enables the model to retain and retrieve long-term patterns across different samples.

**Memory Reading.** Given an input representation $x \in \mathbb{R}^d$ (the temporal-encoded representation $y_{cls}$ from the Transformer encoder), we compute a projected query and retrieve memory content through attention:

$$q = W_{proj} \cdot LN(x) \in \mathbb{R}^{d_m}, \tag{6}$$

$$\boldsymbol{\alpha} = softmax(q\mathcal{M}^\top) \in \mathbb{R}^K, \tag{7}$$

$$r = \boldsymbol{\alpha}\mathcal{M} \in \mathbb{R}^{d_m}, \tag{8}$$

where $\boldsymbol{\alpha}$ represents the similarity distribution between the current brain state and stored memory patterns, and $r$ is the weighted sum of memory slots injecting long-term context into the current representation.

**Memory Writing.** During training, we update memory with an attention-weighted write-in signal and a learnable gate. Denoting the batch-averaged gate and write-in as $\overline{G} \in \mathbb{R}^{K \times d_m}$ and $\overline{W} \in \mathbb{R}^{K \times d_m}$, we update:

$$\mathcal{M} \leftarrow (1 - \overline{G}) \odot \mathcal{M} + \overline{G} \odot \overline{W}, \tag{9}$$

where $\odot$ is element-wise multiplication. This design stabilizes memory evolution and prevents noisy single-sample overwriting while allowing the memory to accumulate recurring patterns.

In the short-term branch, the Brain Function Memory operates on the temporal-encoded representation $y_{cls}$, where $x = y_{cls}$ in the memory reading equation. Since $y_{cls}$ encodes temporal dynamics from positionally-encoded windows $\{z'^{(m)}\}_{m=1}^M$, the memory slots learn to store prototypes of recurring temporal brain dynamics patterns across patients.

### 3.5. GNN-based Long-Term Brain Function Feature Extraction Network

The global adjacency matrix $A \in \mathbb{R}^{N \times N}$ is constructed by applying the same robust precision estimation and thresholding strategy to the entire preprocessed time series, capturing stable long-term connectivity patterns. Let the long-term graph be represented by a weighted adjacency matrix $A$, node 3D coordinates $C \in \mathbb{R}^{N \times 3}$, and community labels $c \in \mathbb{Z}^N$ obtained from modularity maximization on the global graph.

We initialize node features with each node's connectivity profile (row of $A$), degree $d_i = \sum_{j=1}^N |A_{ij}|$, and coordinates, and further inject community information through gated fusion with community embeddings $\text{emb}(c_i)$. We implement message passing using GCN layers.

To stabilize message passing, we add self-loops and construct the symmetric normalized adjacency:

$$D_{ii} = \sum_{j=1}^N |(A + I)_{ij}|, \tag{10}$$

$$\widehat{A} = D^{-1/2}(A + I)D^{-1/2}. \tag{11}$$

Here, $I$ is the identity matrix and $\widehat{A}$ is used for neighborhood aggregation.

Hierarchical message passing is performed with standard residual GCN layers: at each layer, node representations are updated by normalized neighborhood aggregation, a learnable linear transform, and an element-wise nonlinearity, and a residual connection is added for stable optimization.

To obtain a graph-level embedding invariant to node order, we aggregate node representations using parallel mean and max pooling and an attention pooling branch, then concatenate them and project to the final global representation. A memory module with the same read/write mechanism is also employed to store prototypes of recurring static topological patterns.

### 3.6. Feature Fusion

The fusion strategy combines instantaneous dynamical information from the short-term branch with steady-state topological information from the long-term branch.

Each branch employs a two-layer feedforward classifier head with ReLU activation and dropout regularization to enhance nonlinear discriminative capacity.

For the short-term features, the Transformer's CLS output $y^L$, memory readout $r^L$, and projected graph statistics $p_L$ are concatenated as $h^L = [y^L; r^L; p_L]$, and mapped to the branch score by a linear classifier $s^L = w^{L\top} h^L + b^L$.

Similarly, the long-term branch combines the GCN graph embedding $g$, memory readout $r^G$, and projected statistics $p^G$ as $h^G = [g; r^G; p^G]$ and outputs $s^G = w^{G\top} h^G + b^G$.

Decision-level fusion is implemented to retain independent learning capabilities of both branches. We support three fusion strategies: simple averaging $s_{fus} = (s^L + s^G)/2$, weighted linear fusion with learnable weights normalized by softmax, and nonlinear learned fusion through a feedforward network.

The training paradigm maintains discriminative capability of each branch through dual-branch loss. To encourage each branch to learn meaningful representations independently, we employ a training strategy with dynamic weight adjustment. The total training loss is a weighted combination of the two branch-specific task losses,

$$\mathcal{L} = w^L \mathcal{L}_{task}(s^L, y) + w^G \mathcal{L}_{task}(s^G, y), \tag{12}$$

where $y$ is the label and $w^L, w^G$ are branch weights that evolve through a warmup phase with fixed equal weights, a learning phase where weights are jointly optimized with a balance regularization term, and a stable phase where weights are frozen to ensure convergence.

*Table 1.* Performance comparison of 14 different methods on ADHD-200 and OpenNeuro-ds002424 datasets. GTs: Graph Transformers; GNNs: Graph Neural Networks; GTSMs: Graph Temporal Series Methods; LVMs: Large Vision Models. Yellow represents the second baseline model, while green represents the third baseline model. Results are reported as Mean±Std over 4-fold cross-validation with one-sample $t$-test $p$-values against random classifier. Compared to state-of-the-art methods, our model demonstrates excellent detection performance, achieving an average accuracy improvement of 6.97% and an average AUC increase of 10.95% across the two datasets.

| | Methods | ADHD-200 | | | | OpenNeuro-ds002424 | | | |
|---|---|---|---|---|---|---|---|---|---|
| | | ACC (%) | $p$-value | AUC (%) | $p$-value | ACC (%) | $p$-value | AUC (%) | $p$-value |
| GTs | SAN (NeurIPS'21) | 57.45±2.64 | $1.10\times10^{-2}$ | 47.17±2.96 | $1.52\times10^{-1}$ | 57.50±4.66 | $6.86\times10^{-2}$ | 61.20±2.46 | $4.25\times10^{-3}$ |
| | Graph Transformer (AAAI'21) | 57.31±5.02 | $6.19\times10^{-2}$ | 58.23±2.59 | $7.88\times10^{-3}$ | 56.36±8.41 | $2.82\times10^{-1}$ | 63.42±6.88 | $4.31\times10^{-2}$ |
| | Polynormer (ICLR'24) | 60.85±1.45 | $9.92\times10^{-4}$ | 47.02±2.41 | $1.22\times10^{-1}$ | 57.50±0.99 | $6.24\times10^{-4}$ | 54.74±3.85 | $9.07\times10^{-2}$ |
| | BioBGT (ICLR'25) | 56.90±4.01 | $4.12\times10^{-2}$ | 55.74±3.85 | $5.85\times10^{-2}$ | 57.50±3.74 | $2.78\times10^{-2}$ | 53.17±3.69 | $1.84\times10^{-1}$ |
| GNNs | GAT (ICLR'18) | 59.57±2.69 | $5.71\times10^{-3}$ | 55.95±3.04 | $2.96\times10^{-2}$ | 65.91±2.45 | $1.51\times10^{-3}$ | 68.87±0.68 | $1.98\times10^{-5}$ |
| | BrainGNN (MIA'21) | 61.30±0.80 | $1.50\times10^{-4}$ | 51.19±1.34 | $2.22\times10^{-1}$ | 66.82±3.82 | $4.68\times10^{-3}$ | 65.29±4.07 | $7.37\times10^{-3}$ |
| | BrainGB (TMI'23) | 60.19±2.13 | $3.68\times10^{-3}$ | 57.51±3.75 | $4.04\times10^{-2}$ | 54.55±2.23 | $2.66\times10^{-2}$ | 53.39±2.77 | $9.19\times10^{-2}$ |
| | GroupBNA (ICLR'24) | 38.10±7.32 | $6.69\times10^{-2}$ | 49.87±0.13 | $1.82\times10^{-1}$ | 55.00±1.64 | $1.32\times10^{-2}$ | 49.10±0.72 | $1.18\times10^{-1}$ |
| | BrainOOD (ICLR'25) | 60.84±2.57 | $5.30\times10^{-3}$ | 55.38±0.89 | $1.86\times10^{-3}$ | 68.07±4.64 | $6.65\times10^{-3}$ | 66.42±4.27 | $6.90\times10^{-3}$ |
| GTSMs | EBIGNN (JBHI'25) | 61.30±1.09 | $2.45\times10^{-4}$ | 53.62±4.23 | $1.85\times10^{-1}$ | 57.50±7.14 | $1.66\times10^{-1}$ | 57.02±7.28 | $1.93\times10^{-1}$ |
| | ET_MGNN (Neurocomputing'25) | 64.79±2.14 | $4.11\times10^{-4}$ | 65.88±7.18 | $1.07\times10^{-2}$ | 64.24±3.39 | $5.35\times10^{-3}$ | 68.88±9.02 | $3.62\times10^{-2}$ |
| LVMs | Brain Harmony (NeurIPS'25) | 61.04±2.75 | $6.09\times10^{-3}$ | 62.90±2.98 | $4.92\times10^{-3}$ | 54.55±3.21 | $9.17\times10^{-2}$ | 58.50±3.52 | $2.49\times10^{-2}$ |
| | BrainMVP (CVPR'25, Only sMRI) | 52.38±2.15 | $1.14\times10^{-1}$ | 41.34±7.46 | $1.03\times10^{-1}$ | 54.27±5.86 | $2.41\times10^{-1}$ | 51.64±3.45 | $4.12\times10^{-1}$ |
| Our Model | **SLT-BFGN** | **68.62±3.79** | $3.95\times10^{-3}$ | **74.53±1.30** | $7.70\times10^{-5}$ | **78.18±5.51** | $5.18\times10^{-3}$ | **82.13±4.15** | $1.81\times10^{-3}$ |

*Table 2.* 4-fold cross-validation ablation study results on ADHD-200 and OpenNeuro-ds002424 (mean±std over folds). STSE: Short-Term State Encoder; TDE: Temporal Dependency Encoder; BFM: Brain Function Memory; LT: GNN-based Long-Term Network; FF: Feature Fusion. $H_0$: model performance equals random classifier; $p$-values from one-sample $t$-test.

| Components | | | | | ADHD-200 | | | | OpenNeuro | | | |
|---|---|---|---|---|---|---|---|---|---|---|---|---|
| STSE | TDE | BFM | LT | FF | ACC(%) | $p$-value | AUC(%) | $p$-value | ACC(%) | $p$-value | AUC(%) | $p$-value |
| ✓ | × | × | × | × | 65.69±0.69 | $2.32\times10^{-5}$ | 64.65±4.05 | $5.43\times10^{-3}$ | 58.86±2.39 | $5.09\times10^{-3}$ | 50.34±0.84 | $4.70\times10^{-1}$ |
| ✓ | ✓ | × | × | × | 65.83±1.76 | $3.74\times10^{-4}$ | 68.58±8.41 | $2.15\times10^{-2}$ | 58.64±1.58 | $1.63\times10^{-3}$ | 52.89±3.07 | $1.57\times10^{-1}$ |
| ✓ | ✓ | ✓ | × | × | 68.08±1.15 | $7.05\times10^{-5}$ | 73.28±1.58 | $8.51\times10^{-5}$ | 58.63±1.58 | $1.63\times10^{-3}$ | 46.62±1.97 | $4.16\times10^{-2}$ |
| ✓ | ✓ | ✓ | ✓ | × | 65.83±5.93 | $1.28\times10^{-2}$ | 74.71±2.72 | $3.65\times10^{-4}$ | 69.77±5.11 | $4.48\times10^{-3}$ | 78.76±1.81 | $6.89\times10^{-5}$ |
| ✓ | × | ✓ | ✓ | ✓ | 63.83±1.68 | $4.91\times10^{-4}$ | 63.59±3.43 | $4.17\times10^{-3}$ | 74.32±2.27 | $2.23\times10^{-4}$ | 77.03±0.97 | $1.27\times10^{-5}$ |
| ✓ | ✓ | × | ✓ | ✓ | 65.16±3.15 | $2.37\times10^{-3}$ | 73.50±2.74 | $4.33\times10^{-4}$ | 72.28±2.17 | $2.51\times10^{-4}$ | 76.69±3.77 | $7.60\times10^{-4}$ |
| ✓ | ✓ | ✓ | ✓ | ✓ | **68.62±3.79** | $3.95\times10^{-3}$ | **74.53±1.30** | $7.70\times10^{-5}$ | **78.18±5.51** | $5.18\times10^{-3}$ | **82.13±4.15** | $1.81\times10^{-3}$ |

# 4. Experimental Results

## 4.1. Datasets and Experimental Settings

We evaluate our model on two neuroimaging databases: ADHD-200(Bellec et al., 2017) and OpenNeuro-ds002424 (OpenNeuro)(Booth et al., 2020). The ADHD-200 includes data from 357 children with ADHD and 582 controls. OpenNeuro-ds002424 includes multimodal data from 232 ADHD samples and 314 healthy control samples. We use NVIDIA GeForce RTX 4060 GPU for training. The data is split into training (60%), validation (20%), and test (20%) sets, following the hyperparameters of each method for fair comparison. Appendix C presents the methodological details, while Appendix D presents the pseudocode and notation.

To ensure statistical robustness, all experiments are evaluated using 4-fold cross-validation. For each fold, the checkpoint achieving the highest average of ACC and AUC on the validation set is selected for evaluation on the test set.

Per-fold metrics are aggregated to report mean and standard deviation. Statistical significance is assessed via one-sample $t$-tests ($n = 4$ folds, $H_0$: mean performance equals random classifier at 0.5), with $p$-values reported alongside all results.

## 4.2. Overall Performance

As shown in Table 1, SLT-BFGN achieves the highest ACC and AUC on both the ADHD-200 (68.62±3.79% and 74.53±1.30%) and OpenNeuro (78.18±5.51% and 82.13±4.15%) datasets, outperforming the second-best baseline by an average of 6.97% in accuracy and 10.95% in AUC, validating its effectiveness in characterizing short-term neural dynamics and its diagnostic advantage in balancing short-term dynamics and long-term structural features.

Among graph transformer methods, SAN, Graph Transformer, Polynormer, and BioBGT generally perform near chance level on both datasets. Among GNN-based methods, BrainOOD achieves the second-best accuracy on OpenNeuro (68.07±4.64%), yet its AUC on ADHD-200 remains

limited (55.38±0.89%). Graph temporal series methods EBIGNN and ET_MGNN and large vision models Brain Harmony and BrainMVP also show limited performance across both datasets. Overall, SLT-BFGN achieves the best results across all metrics on both datasets.

### 4.3. Ablation Studies

The results of the ablation studies are shown in Table 2. Starting from the short-term state encoder alone, adding the Temporal Dependency Encoder (TDE) improves ADHD-200 AUC from 64.65% to 68.58%, validating the necessity of explicitly modeling sequential brain state transitions beyond window-level spatial encoding alone. Introducing the Brain Function Memory (BFM) further improves ADHD-200 AUC from 68.58% to 73.28% (an absolute increase of 4.70%), the largest single-component gain in the short-term branch, indicating that associating transient activity patterns with cross-subject historical prototypes substantially enhances discrimination. Further adding the Long-Term Graph Network (LT) together with Feature Fusion (FF) leads to a significant performance leap on the OpenNeuro dataset: compared to the model without LT, the complete model achieves ACC 78.18% and AUC 82.13%, indicating that stable structural connectivity features provide important complementary information beyond short-term functional dynamics.

Removing TDE from the complete model causes a notable AUC drop on ADHD-200 (from 74.53% to 63.59%), confirming that sequential transition modeling is the primary driver of temporal discriminability. Removing LT results in a dramatic collapse on OpenNeuro AUC (from 82.13% to 46.62%), demonstrating that long-term structural features are indispensable for cross-site generalization. The complete model achieves the best results across all metrics on both datasets, confirming that the synergistic integration of short- and long-term features comprehensively characterizes the spatiotemporal abnormalities in ADHD brain function.

### 4.4. Temporal Dependency Analysis Based on Graph Sequences

To verify whether temporal order within graph sequences based on short-term windows carries information beyond the spatial connectivity structure itself, we quantitatively assess the independent contribution of temporal order from two complementary perspectives: a temporal shuffling control experiment that directly measures the information loss caused by destroying temporal order, and a Markov ratio analysis that characterizes the structure of temporal dependency. Detailed methodology and results are presented in Appendix A.

**Temporal Shuffling Experiment.** For each sample, the original graph sequence $\mathcal{W} = \{W_1, W_2, \ldots, W_M\}$ is com-

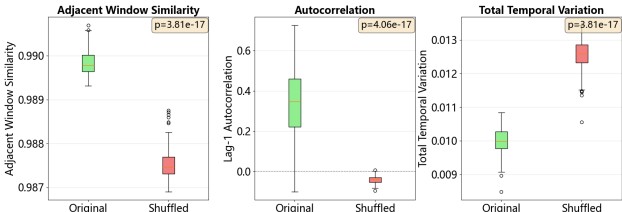

*Figure 4.* Comparison of temporal dependency metrics between original and temporally shuffled graph sequences on the KKI subset. From left to right: adjacent window similarity, lag-1 autocorrelation, and total temporal variation. $p$-values are from paired Wilcoxon signed-rank tests.

pared against a shuffled counterpart $\mathcal{W}^{\text{shuf}}$ generated by randomly permuting the time indices, which fully preserves the spatial connectivity structure and statistical distribution of each window while disrupting only the temporal order. Three complementary metrics are computed for both versions, namely adjacent window similarity, lag-1 autocorrelation, and total temporal variation. The experiment is conducted on the ADHD-200 dataset, with the main analysis performed on the KKI subset (results shown in Figure 4), and replication verification carried out on six additional independent subsets. The original sequence significantly differs from the shuffled sequence across all three metrics. Specifically, the lag-1 autocorrelation remains stably positive across all seven subsets (ranging from 0.276 to 0.384), while it drops close to zero (−0.072 to −0.021) after temporal shuffling, yielding large effect sizes ($d > 0.8$ in all subsets).

**First-Order Markov Property of Brain Functional Connectivity.** To further verify whether the observed temporal dependency conforms to a first-order Markov property, we compute the Markov ratio, defined as the ratio of lag-1 to lag-2 autocorrelation, to directly compare the relative contributions of different time lags to system evolution. As shown in Figure 5, the Markov ratio reaches 4.76 in the KKI subset, significantly greater than 1, confirming that first-order dependency dominates state evolution across short-term windows and that the system's temporal memory is primarily concentrated in the most recent window.

**Neurodynamic Interpretation of the Temporal Structure.** The above results collectively demonstrate that brain functional connectivity states evolve along smooth, constrained trajectories with clear first-order predictability, exhibiting continuity, predictability, and a constrained transition structure. The total temporal variation in the original sequences is approximately 21% lower than shuffled counterparts across all subsets, further confirming that temporal order is not random but carries substantial structural information. These neurodynamic properties directly motivate the design of SLT-BFGN. Specifically, the short-term state encoder captures locally stable connectivity patterns within each win-

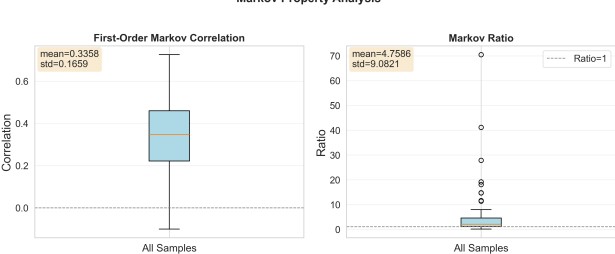

*Figure 5.* First-order Markov correlation (lag-1 autocorrelation) and Markov ratio of original graph sequences on the KKI subset, with mean and standard deviation annotated.

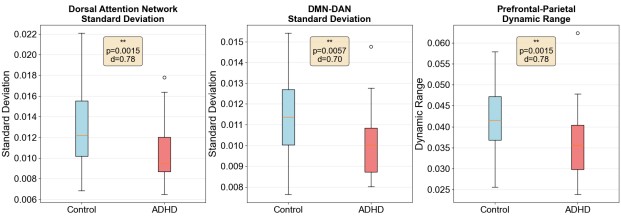

*Figure 6.* Key dynamic abnormalities in ADHD based on network and pathway analysis.

dow, the temporal dependency encoder explicitly models the first-order sequential transitions between windows, and the brain function memory associates current transient patterns with historical prototypes accumulated across subjects.

### 4.5. Dynamic Functional Connectivity Analysis

Based on the verification of significant temporal dependencies in dynamic graph sequences based on short-term windows, we further analyze the abnormalities of ADHD at the level of dynamic functional connectivity patterns from the perspective of neurodynamics. This experiment is based on the fMRI data from the OHSU subset in the ADHD-200 dataset. The analysis is conducted at two scales: the functional network level and the key brain region pathway level, with a focus on the temporal dynamic characteristics of networks related to attention and executive control. For specific details and results, please refer to Appendix B.

The experimental results reveal significant differences in multiple dynamic functional connectivity metrics between the ADHD group and the control group, primarily concentrating in the Dorsal Attention network, inter-network connectivity between the Default Mode network and the Dorsal Attention network, and the prefrontal-parietal pathway, as illustrated in Figure 6. Specifically, the ADHD group generally exhibits reduced dynamic fluctuation amplitude and reduced dynamic range in the aforementioned key networks and pathways, presenting a consistent pattern of limited dynamic flexibility. This result indicates that the functional abnormalities in ADHD are not solely reflected

in the connectivity strength itself, but are more prominently manifested in the weakened ability of functional connectivity to evolve over time.

## 5. Conclusion

In this paper, we address the limitation that current studies pay less attention to short-term brain functional reorganization. An ADHD disease detection model based on short- and long-term brain function encoding and memory graph network is proposed. We exploit a novel brain graph sequence construction method based on short-term windows to extract short-term brain function features. A short-term state and temporal dependency encoder is designed to characterize short-term sequence patterns of brain function. A brain function memory captures associative relationships between neural activity patterns and historical patterns. GNN-based model is introduced to extract long-term structural brain features. These features are fused with short-term features to support ADHD detection. Furthermore, we empirically demonstrate that brain functional connectivity evolution follows a first-order Markov process, providing neurodynamic grounding for the short-term window modeling design. Experimental validation on the publicly available datasets ADHD-200 and OpenNeuro-ds002424 demonstrates the superior performance of our model.

## Acknowledgments

This work is supported by National College Student Innovation Training Program (202510247074).

## Impact Statement

The proposed SLT-BFGN model establishes a new framework for ADHD detection by jointly modeling short-term brain functional reorganization and long-term structural connectivity. Its adoption could advance clinical diagnosis of neuropsychiatric disorders and inspire future graph-based approaches for analyzing dynamic brain networks. As with any medical AI system, deployment would require careful validation and attention to issues of data diversity, interpretability, and equitable access; this work contributes toward greater transparency in modeling brain dynamics, which may help address some of those concerns.

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

# A. Temporal Dependency Analysis for Brain Graph Sequences Based on Short-Term Windows

## A.1. Temporal Dependency Analysis

To verify whether the temporal order in the sequence of dynamic functional connectivity graphs based on short-term windows carries effective information beyond the spatial connectivity structure itself, we design a rigorous temporal dependency analysis framework. This framework compares the original graph sequence with its temporally shuffled version to quantitatively assess the contribution of temporal order in the dynamics of functional connectivity.

We design a temporal shuffling control. For each sample, we extract the complete graph sequence from resting-state fMRI data, represented as a set of adjacency matrices varying over time:

$$\mathcal{W} = \{W_1, W_2, \ldots, W_M\}, \quad W_m \in \mathbb{R}^{N \times N}, \tag{13}$$

where $M$ is the number of time windows, and $N$ is the number of brain region nodes. Each $W_m$ represents the functional connectivity graph estimated within the $m$-th time window. The temporal shuffling operation generates a shuffled sequence by randomly permuting the time indices:

$$\mathcal{W}^{\text{shuf}} = \{W_{\pi(1)}, W_{\pi(2)}, \ldots, W_{\pi(M)}\}, \tag{14}$$

where $\pi(\cdot)$ denotes a random permutation. This operation fully retains the spatial connectivity structure and its statistical distribution within each time window, while only disrupting the temporal order between different time windows. Therefore, any systematic differences between the original and shuffled sequences can be explicitly attributed to the loss of temporal order information across short-term windows.

### A.1.1. TEMPORAL DEPENDENCY METRICS

We characterize the temporal structure of the graph sequence from three complementary perspectives, reflecting state continuity, short-term memory, and the overall degree of dynamic constraints.

**(1) Adjacent Window Similarity**

The similarity between adjacent time windows is used to measure the continuity of connectivity states as they evolve over time. For two consecutive adjacency matrices $W_m$ and $W_{m+1}$, we first calculate the Frobenius norm difference:

$$D_m = \|W_{m+1} - W_m\|_F, \tag{15}$$

and normalize it by the maximum possible difference:

$$\tilde{D}_m = \frac{D_m}{\sqrt{2N^2}}. \tag{16}$$

Based on this, the similarity is defined as:

$$S_m = \frac{1}{1 + \tilde{D}_m}, \tag{17}$$

ensuring that the similarity values range from $(0, 1]$, with higher values indicating greater similarity between adjacent time windows. The final adjacent window similarity metric is defined as the average over the entire sequence:

$$\text{Similarity} = \frac{1}{M-1} \sum_{m=1}^{M-1} S_m. \tag{18}$$

This metric directly reflects the smooth temporal evolution of functional connectivity states: if the connectivity patterns change dramatically between adjacent time windows, the similarity will significantly decrease.

**(2) Lag-1 Autocorrelation**

To characterize whether connectivity states exhibit short-term memory across successive short-term windows, we calculate the lag-1 autocorrelation of the graph sequence. Specifically, for each time window, we first flatten the adjacency matrix into a vector and compute the global average edge weight:

$$g_m = \frac{1}{N^2} \sum_{i,j} W_m(i,j), \tag{19}$$

resulting in a one-dimensional time series $\{g_1, g_2, \ldots, g_M\}$. The lag-1 autocorrelation coefficient is defined as:

$$\rho_1 = \text{corr}\big(\{g_m\}_{m=1}^{M-1}, \{g_{m+1}\}_{m=1}^{M-1}\big), \tag{20}$$

where $\text{corr}(\cdot, \cdot)$ denotes the Pearson correlation coefficient. If $\rho_1 > 0$, it indicates that the overall connectivity strength at the current time has predictive power over the next time step, reflecting a typical short-term dependency structure. Conversely, if the sequence is approximately random in time, this autocorrelation coefficient should be close to zero.

To further verify whether this temporal dependency conforms to a first-order Markov property, we also compute the second-order (lag-2) and third-order (lag-3) autocorrelation coefficients:

$$\rho_2 = \text{corr}\big(\{g_m\}_{m=1}^{M-2}, \{g_{m+2}\}_{m=1}^{M-2}\big), \quad \rho_3 = \text{corr}\big(\{g_m\}_{m=1}^{M-3}, \{g_{m+3}\}_{m=1}^{M-3}\big), \tag{21}$$

and examine their decay with increasing lag order. If the correlation decays rapidly from first-order to third-order, it suggests that the system's temporal memory is primarily concentrated in the most recent short-term window, consistent with the characteristics of a first-order Markov process.

### (3) Total Temporal Variation

To quantify the overall fluctuation amplitude of the entire graph sequence in the temporal dimension, we define the total temporal variation metric as the cumulative absolute difference between adjacent time windows:

$$TV = \sum_{m=1}^{M-1} \sum_{i<j} |W_{m+1}(i,j) - W_m(i,j)|. \tag{22}$$

Considering differences in the number of time windows and node scales across samples, we further normalize this quantity:

$$TV_{\text{norm}} = \frac{TV}{(M-1) \cdot \frac{N(N-1)}{2}}, \tag{23}$$

where $M-1$ represents the number of adjacent time window pairs, and $\frac{N(N-1)}{2}$ denotes the number of possible edge pairs in an undirected graph. This metric reflects the overall change amplitude of connectivity patterns over time, with higher values indicating greater instability in connectivity states along the temporal dimension.

In addition to the three temporal dependency metrics above, we further introduce two metrics directly related to Markov properties to enhance the discriminative ability of temporal dependency across short-term windows.

### (4) First-Order Markov Correlation

This metric is numerically equivalent to the lag-1 autocorrelation $\rho_1$, but in the context of Markov property testing, its significance lies in evaluating the predictive ability of the current state for the state at the next window. If $\rho_1$ is significantly positive, it indicates that the current connectivity pattern $W_m$ contains the key information needed to predict the next state $W_{m+1}$, which is a necessary condition for the first-order Markov assumption to hold.

### (5) Markov Ratio

To directly compare the relative contributions of different time lags to system evolution, we define the Markov ratio as the ratio of the absolute value of the lag-1 (first-order) to the lag-2 (second-order) autocorrelation:

$$R = \frac{|\rho_1|}{|\rho_2| + \epsilon}, \tag{24}$$

where $\epsilon = 10^{-6}$ is used to prevent division by zero. This ratio is used to test the first-order Markov assumption across short-term windows. When $R > 1$, it indicates that the current window state has a stronger influence on the next window than on more distant ones, consistent with a first-order Markov process; if $R \approx 1$, higher-order memory structures may be required; if $R < 1$, it suggests significant long-range temporal dependencies in the system. For a typical first-order Markov process, this ratio is expected to be significantly greater than 1.

For each sample, we calculate the values of the three metrics for both the original sequence and the temporally shuffled sequence. The statistical comparison employs a paired design to eliminate the influence of inter-individual differences. Specifically, we use paired $t$-tests and Wilcoxon signed-rank tests to assess the statistical significance of the differences. The $p$-values of the Wilcoxon signed-rank test are reported in the following experiments, with the significance level set at $\alpha = 0.05$.

To quantify the magnitude of the differences, we report the Cohen's $d$ effect size for paired samples, defined as:

$$d = \frac{\mu_\Delta}{\sigma_\Delta}, \tag{25}$$

where $\Delta = X_{\text{orig}} - X_{\text{shuf}}$ represents the metric difference between the original and shuffled sequences for each sample, and $\mu_\Delta$ and $\sigma_\Delta$ are the mean and standard deviation of the differences, respectively. According to the rule of thumb, $|d| < 0.2$ indicates a small effect; $|d| \approx 0.5$ indicates a medium effect; $|d| > 0.8$ indicates a large effect. Compared to the $p$-value alone, Cohen's $d$ directly reflects the discriminative strength contributed by temporal order at the individual level, providing a more meaningful practical interpretation.

### A.1.3. MULTI-SUBSET VALIDATION

To validate the robustness of temporal dependency across diverse scanning sites and populations, we repeat the temporal shuffling experiment on six additional independent subsets from the ADHD-200 dataset (NeuroIMAGE, NYU, OHSU, Peking, Pittsburgh, and WashU).

## A.2. Results in the KKI Subset

**Temporal Dependency Analysis.** We first conduct a temporal dependency analysis on the original resting-state functional connectivity graph sequences from the KKI subset ($n = 94$) to characterize the temporal evolution of brain connectivity states. The three complementary metrics consistently and significantly reveal temporal structure at the group level.

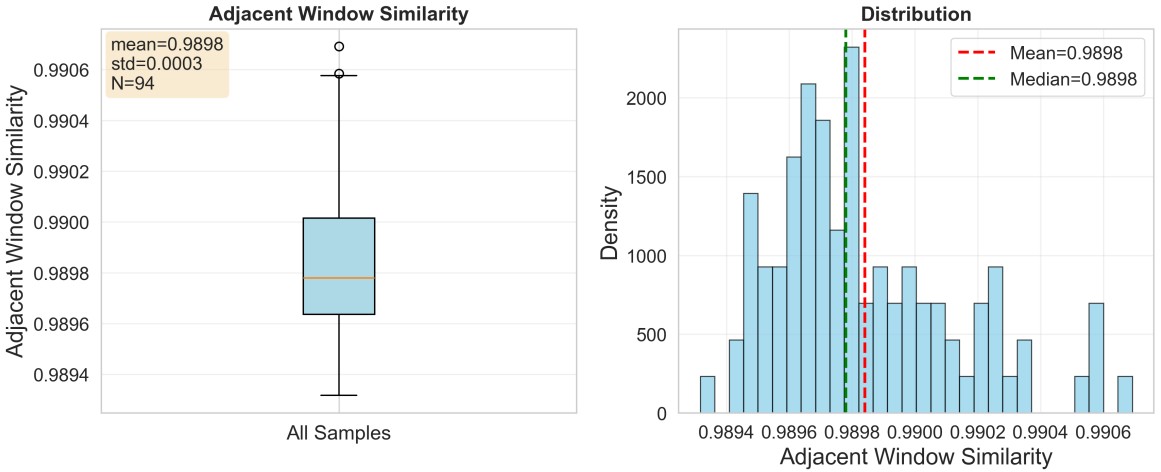

*Figure 7.* Adjacent window similarity of samples in the KKI subset.

In terms of adjacent window similarity, all samples exhibit extremely high continuity, with an average similarity of 0.9898±0.0003, close to the theoretical upper limit of 1.0, as shown in Figure 7. This indicates that the brain network states within a relatively short term remain almost unchanged in overall structure. This result implies that the evolution of functional connectivity has extremely strong temporal continuity rather than random jumps in the state space. It is important to emphasize that this high similarity is not an analytical flaw but rather a key piece of evidence for the rationality of dynamic functional connectivity: if adjacent time windows were completely unrelated, it would more likely reflect noise dominance or improper window settings.

This finding is highly consistent with existing dynamic functional connectivity studies. Allen et al. (Allen et al., 2014) systematically analyzed whole-brain functional connectivity dynamics using a sliding window approach, finding that dynamic

connectivity patterns between adjacent time windows exhibited significant continuity, reflecting smooth state transitions rather than abrupt changes. Hutchison et al. (Hutchison et al., 2013) also systematically summarized this phenomenon, arguing that high similarity between adjacent windows is an important metric in dynamic functional connectivity analysis that reflects true neural dynamics rather than random fluctuations. Therefore, the observed adjacent window similarity of 0.9898±0.0003 in this study indicates that the extracted functional connectivity graph sequence based on short-term windows exhibits good temporal continuity, reflecting the smooth evolutionary trajectory of brain functional states.

The lag-1 autocorrelation analysis further reveals the temporal dependency between connectivity states. The lag-1 auto-correlation coefficient in the original sequence is 0.336±0.166 and rapidly decays with increasing lag order, approaching zero (−0.004±0.227) by lag-2, as shown in Figure 8. Lag-3 further decreases to −0.039±0.222. This pattern indicates that the functional connectivity state at the current window has a moderate predictive power over the next time step, but the dependency on earlier historical states rapidly diminishes, consistent with the typical characteristics of a first-order Markov process. In the existing literature, Liégeois et al. (Liégeois et al., 2019) systematically analyzed the temporal structure of resting-state dynamic functional connectivity, reporting that lag-1 autocorrelation in healthy subjects was significantly positive while higher-order lags rapidly decayed. The value of 0.336±0.166 obtained in this study indicates that the observed temporal dependency has good physiological plausibility.

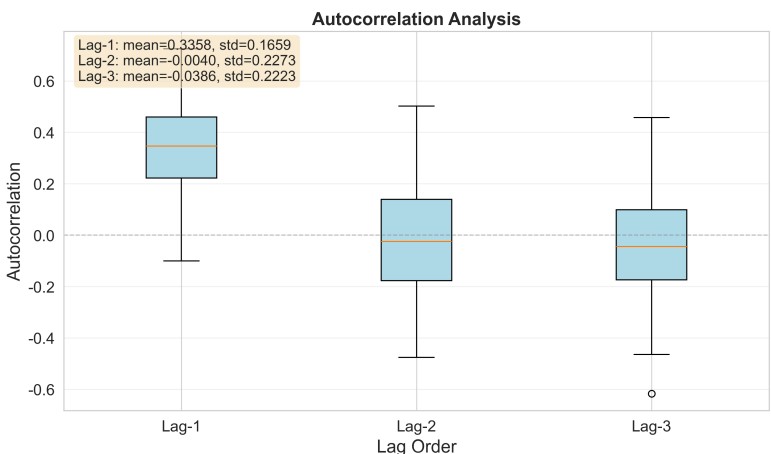

*Figure 8.* Lag-1, lag-2, and lag-3 autocorrelation of samples in the KKI subset.

The total temporal variation metric reflects the overall fluctuation range of connectivity dynamics. As shown in Figure 9, the average total variation in the original sequence is 0.0100±0.0004, with minimal inter-individual differences, indicating that although connectivity states change over time, the amplitude of these changes is strictly constrained. This result supports the view that the evolution of functional connectivity across short-term windows is smooth and stable at the system level.

Additionally, the Markov ratio reaches 4.76, significantly larger than 1, as shown in Figure 5 in the main text, indicating that first-order dependency dominates state evolution across short-term windows, i.e., $W_{m+1}$ is primarily determined by $W_m$ rather than relying on earlier historical states. Although this ratio is an original measure proposed in this study, its theoretical foundation can be traced back to the classical Hidden Markov Model (HMM) framework. Rabiner (Rabiner, 1989) noted that the first-order Markov assumption is the most fundamental and stable form in sequence modeling. When higher-order dependencies contribute significantly less than first-order dependencies, the first-order model is not only theoretically valid but also computationally advantageous. Furthermore, Vidaurre et al. (Vidaurre et al., 2017) analyzed resting-state fMRI data from 820 subjects using hidden Markov models, finding that brain network state transitions exhibited significant non-randomness, with current states having strong predictive power over the next state, which highlights the core role of the first-order transition matrix in describing brain network dynamics.

In summary, the KKI subset data exhibit three core temporal characteristics of resting-state functional connectivity: high continuity, strong temporal dependency, and low overall fluctuation. However, these observations alone cannot rule out the possibility that the above characteristics may simply arise from the temporal smoothness of fMRI signals or the slow nature of hemodynamic responses, rather than genuine temporal information. Therefore, it is necessary to further validate these findings through control experiments.

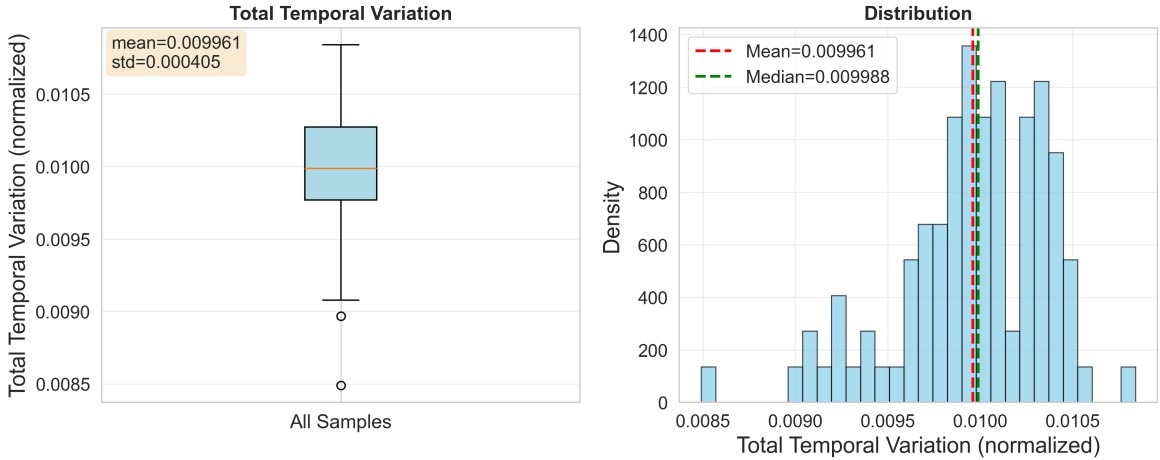

*Figure 9.* Total temporal variation of samples in the KKI subset.

**Temporal Shuffling Experiment: Separating Temporal Information from Spatial Structure.** To test whether temporal order within the sequence of short-term windows carries additional information independent of spatial connectivity structure, we perform random temporal shuffling (100 times) on the connectivity sequences of each sample. This process, which only disrupts the temporal order while fully preserving the spatial structure and statistical distribution of each time window, leads to the recalculation of the three metrics mentioned above.

The impact of temporal shuffling is extremely strong and consistent across all three metrics. The adjacent window similarity drops to 0.9876±0.0004 after shuffling, with an absolute difference of only 0.0022, but the effect size is as high as Cohen's $d = 11.04$ ($p < 10^{-16}$), and all 94 samples show that the original sequence is higher than the shuffled sequence, as shown in Figure 10. Considering that the original similarity is already close to the theoretical upper limit, this difference represents a significant change in the remaining variable space, clearly indicating that high continuity is not merely caused by signal smoothing.

The lag-1 autocorrelation decreases sharply from 0.336±0.166 to $-0.042$±0.018 after shuffling, approaching zero, which is the expected baseline for a completely random sequence, with an effect size of $d = 2.27$ ($p < 10^{-16}$), as shown in Figure 18. This result indicates that the predictability of states in the original sequence entirely depends on the correct temporal order, which is completely destroyed once shuffled, directly proving the temporal dependency of functional connectivity evolution.

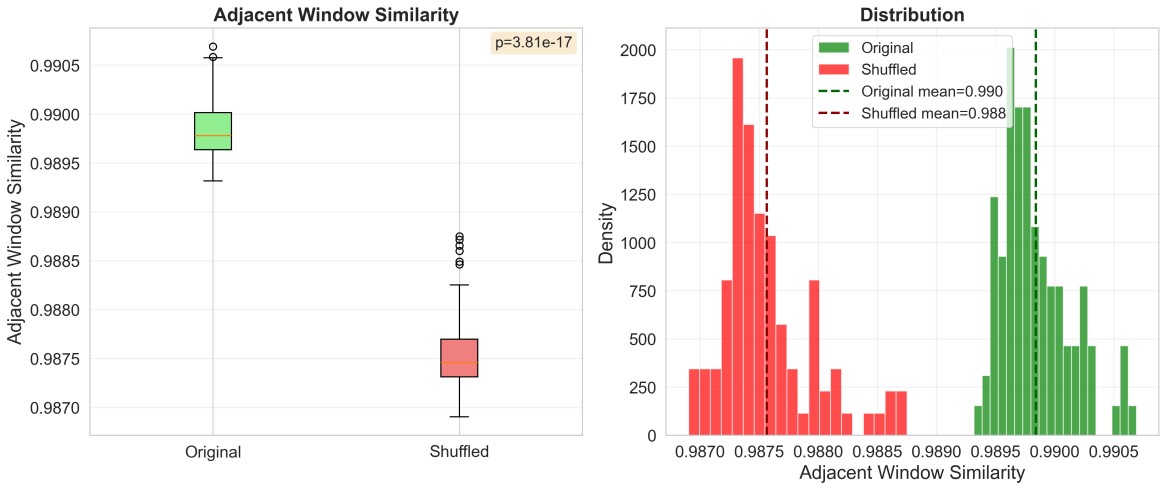

*Figure 10.* Comparison of adjacent window similarity of samples before and after shuffling in the KKI subset.

Meanwhile, the total temporal variation significantly increases to 0.0125±0.0005 after shuffling ($d = -10.75, p < 10^{-16}$),

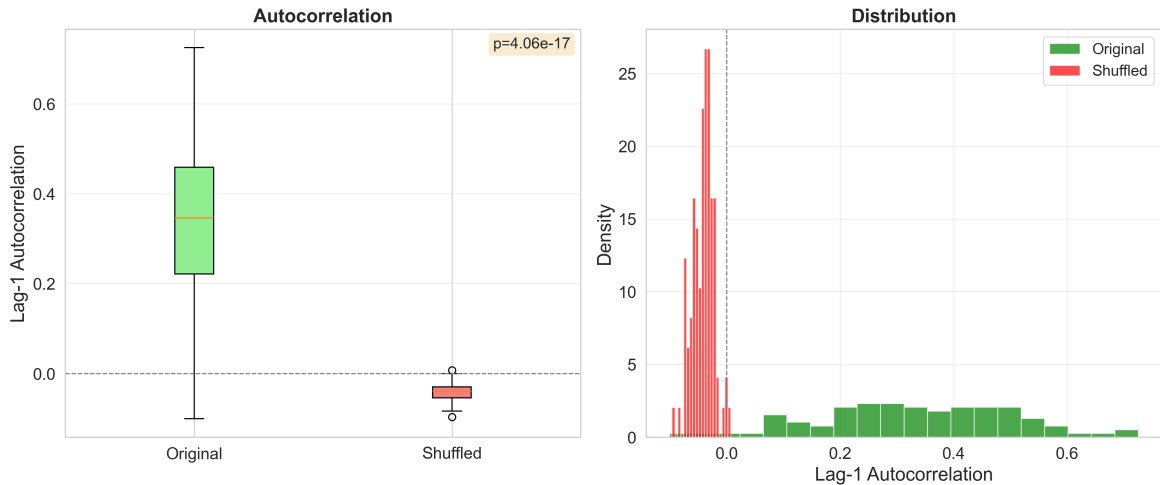

*Figure 11.* Comparison of lag-1 autocorrelation of samples before and after shuffling in the KKI subset.

as shown in Figure 19, indicating that the original temporal order constrains state transitions, resulting in smoother connectivity changes compared to those occurring randomly.

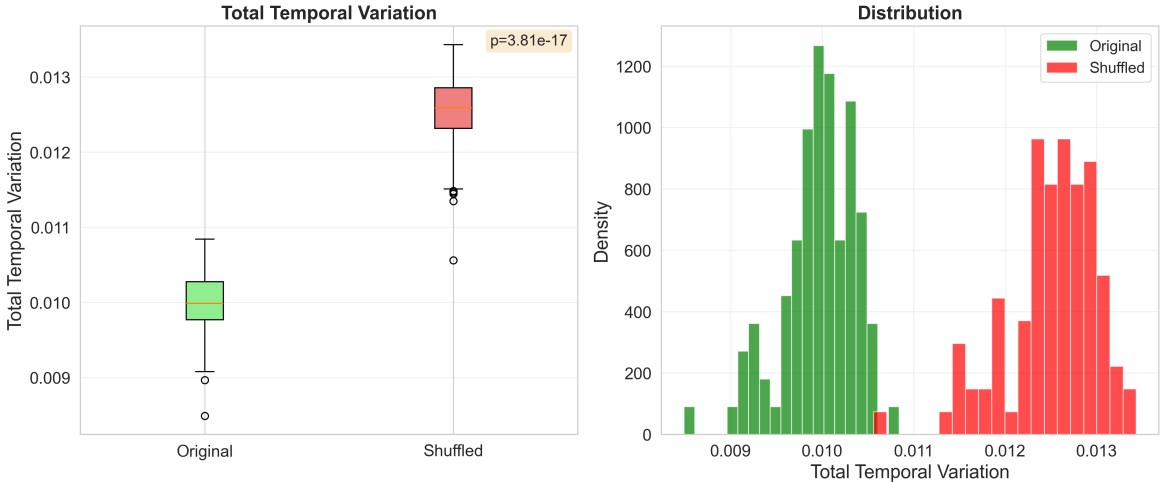

*Figure 12.* Comparison of total temporal variation of samples before and after shuffling in the KKI subset.

In summary, the temporal shuffling experiment clearly demonstrates that the temporal order of brain functional connectivity graph sequences carries a substantial amount of information that cannot be explained by spatial structure alone. Temporal order encodes the continuous transition of connectivity states, the predictive relationships between states, and the permissible range of transitions, with overall characteristics consistent with the assumption of a first-order Markov process. These findings provide a solid theoretical basis for subsequent analyses of dynamic functional connectivity across short-term windows.

### A.3. Validation Across Other ADHD-200 Subsets

To test whether temporal dependency is a universal feature of human brain functional connectivity, we repeat the same temporal shuffling analysis on six remaining independent ADHD-200 subsets (NeuroIMAGE, NYU, OHSU, Peking, Pittsburgh, WashU). Including the KKI subset, the total sample size is expanded to 941 samples, covering diverse geographical regions, scanning protocols, and subject populations.

As shown in Figure 13, all seven subsets exhibit a highly consistent pattern of temporal dependency. In terms of adjacent window similarity, the original sequences of all subsets are significantly higher than the shuffled sequences, with Cohen's $d$

effect sizes ranging from 5.51 (NYU) to 11.96 (Peking), all indicating extremely large effects ($d > 5$).

The multi-subset results of lag-1 autocorrelation further validate the universal presence of the Markov property. All subsets exhibit stable positive autocorrelation in the original sequences (ranging from 0.276 to 0.384), while autocorrelation approaches zero after temporal shuffling ($-0.072$ to $-0.021$). The Cohen's $d$ effect sizes range from 1.48 (OHSU) to 3.00 (NeuroIMAGE), with all subsets exhibiting large effects ($d > 0.8$). Notably, the KKI ($d = 2.27$) and NeuroIMAGE ($d = 3.00$) subsets show particularly significant effects, indicating that the predictability of the current connectivity state over the next state is not sporadic but stably present across different populations.

The total temporal variation results across all subsets also support the conclusion that connectivity state transitions are constrained by temporal order. The original sequences of all subsets exhibit significantly lower variation than the shuffled sequences, averaging about 21% lower, with Cohen's $d$ effect sizes ranging from $-5.00$ (NYU) to $-10.75$ (KKI). This effect is particularly pronounced in the KKI ($d = -10.75$) and Peking ($d = -10.41$) subsets, indicating that real brain connectivity dynamics cover only a restricted trajectory in the state space rather than all possible random transitions.

Overall, the results across seven independent subsets indicate that temporal dependency and its manifestation of the first-order Markov property are robust characteristics of human resting-state functional connectivity, rather than artifacts of specific subsets, scanning devices, or sample compositions. From a methodological perspective, this result provides a clear benchmark for subsequent dynamic connectivity modeling: models that can effectively utilize temporal information should stably distinguish between original and temporally shuffled sequences across subsets and achieve effect sizes comparable to those in this study.

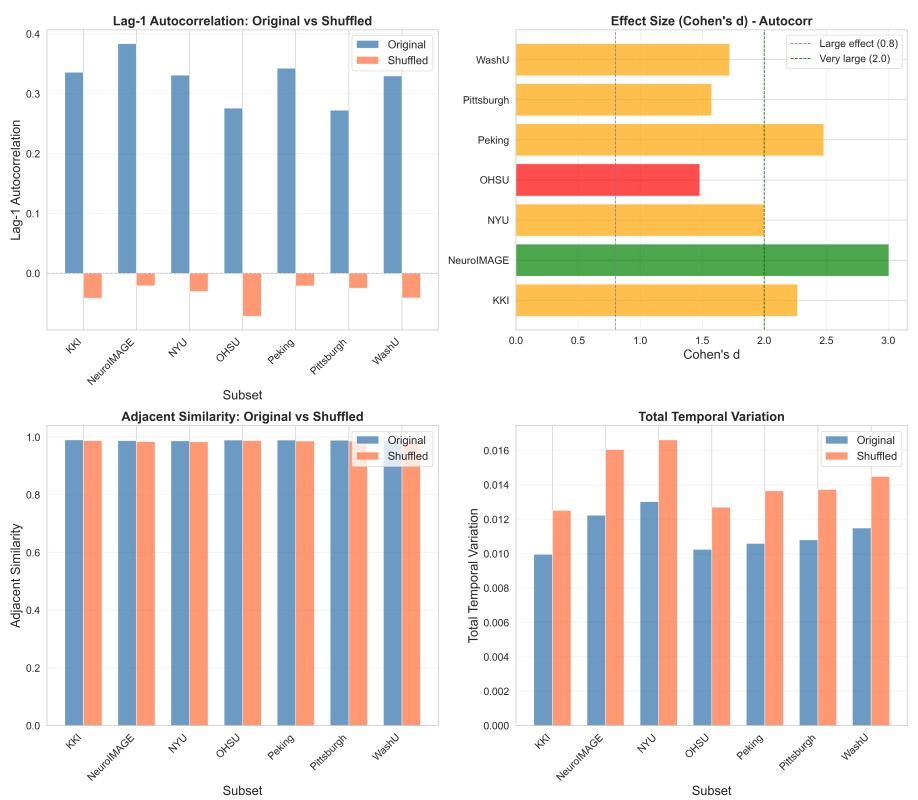

*Figure 13.* Comparison of temporal dependency metrics across subsets of the ADHD-200 dataset.

## A.4. The Theoretical Significance of Markov Properties

The consistent temporal dependency patterns observed across subsets provide direct empirical support for the notion that brain functional connectivity evolution approximately follows a first-order Markov process. The core assumption of a first-order Markov process is that the next state of the system depends only on the current state and does not require longer historical information. This characteristic is systematically reflected in the three temporal dependency metrics of this study.

Firstly, the significantly higher adjacent window similarity compared to the shuffled sequences indicates that adjacent connectivity states are not randomly paired but evolve smoothly along continuous, structured trajectories. If functional connectivity states could freely jump in the state space, the similarity between adjacent states should not be significantly higher than that of randomly shuffled sequences. The observed large effect size suggests that temporal order plays a decisive role in the organization of connectivity states.

Secondly, the stable positive lag-1 autocorrelation across subsets directly reflects the Markov property. The significantly positive autocorrelation in the original sequences implies that the current connectivity state has clear predictive power for the next state, which rapidly disappears after temporal shuffling. This result indicates that the dynamic evolution of brain functional connectivity is not composed of independent instantaneous states but is a state transition process with memory.

Thirdly, the significantly lower total temporal variation in the original sequences compared to the shuffled sequences (averaging about 21% lower) reveals the constraints on state transitions. Under the Markov framework, the system typically evolves through a limited set of state transitions rather than freely jumping between arbitrary states. Our results indicate that the actual evolutionary path of brain functional connectivity covers only a small part of the theoretical state space, reflecting the combined effects of physiological, metabolic, and structural connectivity constraints.

From a theoretical perspective, these findings together characterize the fundamental features of brain functional connectivity dynamics: continuity, predictability, and constraint. These features imply that the evolution of functional connectivity can be regarded as a self-organizing, homeostatically regulated dynamical process, where each moment's state inherits from the previous state and sets constraints for subsequent evolution. This description is highly consistent with the theoretical framework of viewing the brain as a predictive system, where the system must have sufficient temporal predictability to achieve efficient information processing and state regulation.

More importantly, this Markov property provides a clear theoretical basis for subsequent methodological choices. Traditional dynamic functional connectivity analyses often treat different time windows as independent, ignoring the sequential relationships between states, thereby losing a large amount of temporal information. In contrast, adopting the graph sequence method can better capture the predictability of the current state over the next state. The effect sizes observed in this study indicate that the temporal dimension is not a weak feature but carries information content as important as spatial structure.

Therefore, the use of sequential modeling based on short-term windows to analyze dynamic functional connectivity is not merely a technical choice but is determined by the intrinsic dynamical properties of brain connectivity evolution. Although sequential models have higher computational complexity than static methods, they can systematically utilize temporal dependencies and potentially bring significant information gains.

## B. Dynamic Connectivity Pattern Analysis for Brain Graph Sequences Based on Short-Term Windows

### B.1. Dynamic Connectivity Pattern Analysis

After confirming the significant temporal dependency in the dynamic functional connectivity sequences based on short-term windows, we further investigate whether ADHD patients exhibit abnormalities in the dynamic functional connectivity patterns. This section is based on data from the OHSU subset, with 70 control subjects and 43 ADHD patients. After strict data quality control, the final effective samples include 45 controls and 26 ADHD patients.

To avoid potentially masking local pathological information through whole-brain average analysis, we conduct a fine-grained characterization of ADHD-related local dynamic abnormalities from two aspects: the internal dynamics of functional networks and the dynamic connectivity features between key brain regions.

At the functional network level, we employ the 7-network functional partitioning scheme proposed by Yeo et al. (Thomas Yeo et al., 2011), dividing the whole brain into Visual, Somatomotor, Dorsal Attention, Ventral Attention, Limbic, Frontoparietal, and Default Mode networks. This partitioning scheme has good biological interpretability in cognitive and psychiatric research and helps identify ADHD-related local dynamic abnormalities at the functional module scale. Unlike traditional whole-brain coarse-grained metrics, this modular analysis can reveal potential pathological changes within specific functional networks even when no significant differences are observed at the overall level.

In this study, the 116 AAL ROIs are approximately assigned to the Yeo 7 functional networks based on their anatomical locations and reported functional affiliations in the literature, as shown in Figure 14. Since AAL is an anatomical atlas and

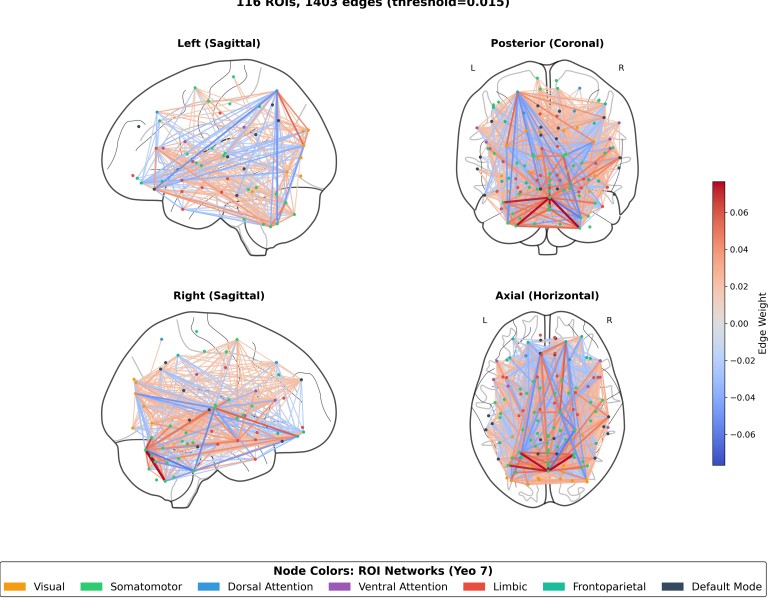

*Figure 14.* Distribution of functional networks.

Yeo is a functional atlas, this mapping is approximate, especially for brain regions with ambiguous functional boundaries (e.g., the junction between the Frontoparietal and Dorsal Attention networks), where classification uncertainty may exist.

For the dynamic features within networks, we calculate the relative dynamic variability and absolute dynamic amplitude of connections within each functional network. Specifically, for the $k$-th functional network, let $E_k$ be the set of all edges $(i, j)$ within the network. We define the coefficient of variation (CV) of connections within the network as

$$\text{CV}_k = \frac{1}{|E_k|} \sum_{(i,j) \in E_k} \frac{\sigma_{ij}}{|\mu_{ij}| + \epsilon}, \tag{26}$$

where $\sigma_{ij}$ and $\mu_{ij}$ represent the standard deviation and mean of edge $(i, j)$ over time, respectively, and $\epsilon = 10^{-8}$ is used to prevent division by zero. The coefficient of variation reflects the dynamic instability of connection strengths in a normalized sense, with higher values indicating greater fluctuations within the network.

Additionally, we calculate the within-network standard deviation (STD)

$$\text{STD}_k = \frac{1}{|E_k|} \sum_{(i,j) \in E_k} \sigma_{ij}, \tag{27}$$

to characterize the absolute amplitude of connection strength changes over time. CV and STD describe the dynamic characteristics within functional networks from relative and absolute perspectives, respectively, complementing each other.

To further characterize the dynamic coordination between different functional networks, we define an inter-network dynamic variability metric. For any two functional networks $A$ and $B$, let $E_{A-B}$ be the set of all inter-network connections between them.

The absolute dynamic fluctuations of inter-network connections are characterized by the inter-network standard deviation metric, defined as:

$$\text{STD}(A - B) = \frac{1}{|E_{A-B}|} \sum_{(i,j) \in E_{A-B}} \sigma_{ij}. \tag{28}$$

Here, $\sigma_{ij}$ represents the standard deviation of the inter-network connection $(i, j)$ over time. This metric reflects the overall fluctuation amplitude of functional coupling strength between two functional networks, with higher values indicating lower stability of inter-network dynamic coordination.

To characterize the relative dynamic instability of inter-network connections in a normalized sense, we further define the inter-network coefficient of variation metric:

$$\text{CV}(A - B) = \frac{1}{|E_{A-B}|} \sum_{(i,j)\in E_{A-B}} \frac{\sigma_{ij}}{|\mu_{ij}| + \epsilon}, \tag{29}$$

where $\mu_{ij}$ represents the mean of the inter-network connection $(i, j)$ over time, and $\epsilon = 10^{-8}$ is used to prevent division by zero. This metric reflects the dynamic fluctuation of inter-network functional connections relative to their average strength, with higher values indicating more unstable dynamic coupling between the two functional networks.

At the key brain region pairs level, we focus on three core functional pathways repeatedly reported in ADHD neuropathology studies: the prefrontal-striatal pathway (involved in reward processing and inhibitory control), the prefrontal-parietal pathway (closely related to executive function and attention control), and the anterior cingulate-amygdala pathway (involved in emotion regulation). For any pair of brain regions $(R_1, R_2)$, let $E$ be the set of all connections between them. We calculate two types of dynamic metrics:

**(1) Dynamic Range (Range)**: Defined as the average of the differences between the maximum and minimum values of each connection over the time series:

$$\text{Range} = \frac{1}{|E|} \sum_{(i,j)\in E} \left( \max_t x_{ij}(t) - \min_t x_{ij}(t) \right), \tag{30}$$

where $x_{ij}(t)$ represents the connection strength between regions $i$ and $j$ at time $t$. This metric reflects the overall fluctuation range of connection strengths.

**(2) Maximum Standard Deviation (MaxSTD)**: Defined as the maximum temporal standard deviation among all connections in the brain region pair:

$$\text{MaxSTD} = \max_{(i,j)\in E} \sigma_{ij}, \tag{31}$$

where $\sigma_{ij}$ is the standard deviation of the connection strength $x_{ij}(t)$ over time. This metric is particularly sensitive to extreme or sudden dynamic changes and helps identify potential abnormal connections.

**Statistical Analysis.** We employ the Mann-Whitney U test to compare the control and ADHD groups. The null hypothesis $H_0$ assumes no significant difference in the distribution of the metric between the two groups, while the alternative hypothesis $H_1$ posits a significant difference. The significance level is set at $\alpha = 0.05$. Additionally, Cohen's $d$ effect size is calculated to quantify the magnitude of the inter-group difference, defined as $d = (\bar{x}_{\text{Control}} - \bar{x}_{\text{ADHD}})/s_{\text{pooled}}$, where $s_{\text{pooled}}$ represents the pooled standard deviation. Unlike the Cohen's $d$ for paired samples in Appendix A, the effect size here uses the independent sample version.

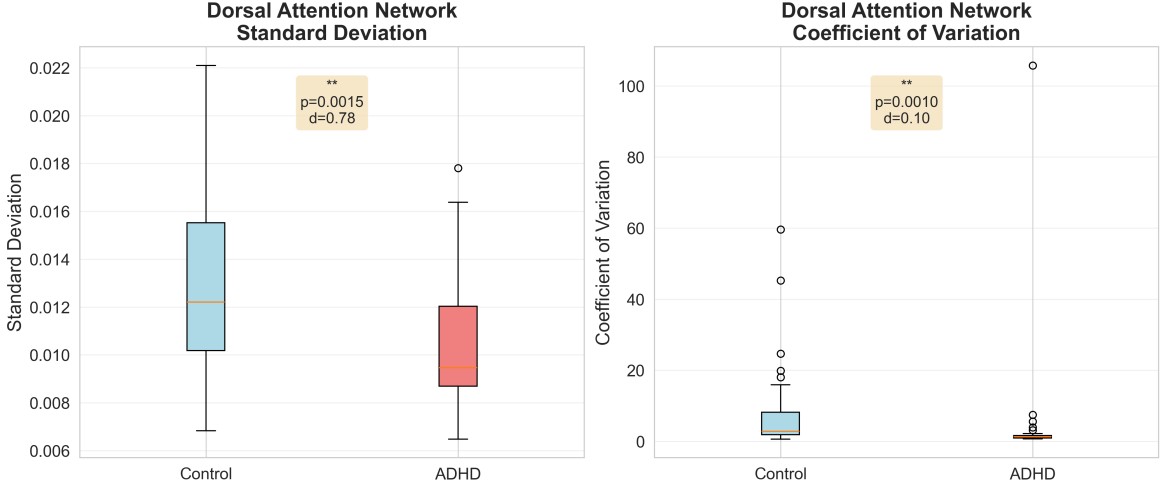

*Figure 15.* Within-network dynamic differences between ADHD and control groups.

## B.2. Results in the OHSU Subset

In the OHSU subset data, after performing group-wise statistical tests on 31 refined dynamic functional connectivity metrics, a total of 6 metrics reached the significance level (significance rate 19.4%), significantly higher than the expected number of significant metrics under random conditions at the 5% significance level (approximately 1.55). These significant metrics were primarily distributed in the Dorsal Attention Network (DAN), inter-network connections between the Default Mode Network and the DAN, and the prefrontal-parietal pathway, collectively presenting a dynamic abnormality pattern centered around the attention control system.

**(1) Significant abnormalities in the internal dynamic characteristics of the Dorsal Attention Network (DAN).** As shown in Figure 15, within this network, the temporal standard deviation of internal connections exhibited a significant difference between groups ($p = 0.0015$, Cohen's $d = 0.782$), representing the largest effect size among all significant metrics. The Dorsal Attention Network primarily supports top-down goal-directed attention control, with key nodes including the Frontal Eye Field (FEF, corresponding to the superior frontal gyrus) and the Intraparietal Sulcus (IPS, corresponding to the superior parietal lobule), which are crucial neural foundations for maintaining sustained attention and spatial orientation. The significant alteration in the dynamic amplitude of connections within this network indicates that the attention system's functional state is more unstable over time, making it difficult to maintain a stable and consistent activation pattern. This characteristic is highly consistent with the clinical manifestation of sustained attention difficulties in ADHD patients, where attention states frequently fluctuate between different time periods and cannot be maintained at a task-relevant level for long. Unlike the orderly regulation based on task demands in healthy individuals, these fluctuations are more likely to reflect disordered neural noise interference. Additionally, the coefficient of variation within the Dorsal Attention Network also reached a significant level ($p = 0.0010$, $d = 0.096$), further validating the presence of dynamic abnormalities within the network from the perspective of relative dynamic fluctuations. This dynamic instability provides a direct neurodynamic explanation for the sustained attention difficulties in ADHD patients, consistent with the dual-system model of attention proposed by Corbetta and Shulman, which emphasizes the central role of the dorsal attention network in maintaining goal-directed attention (Corbetta & Shulman, 2002).

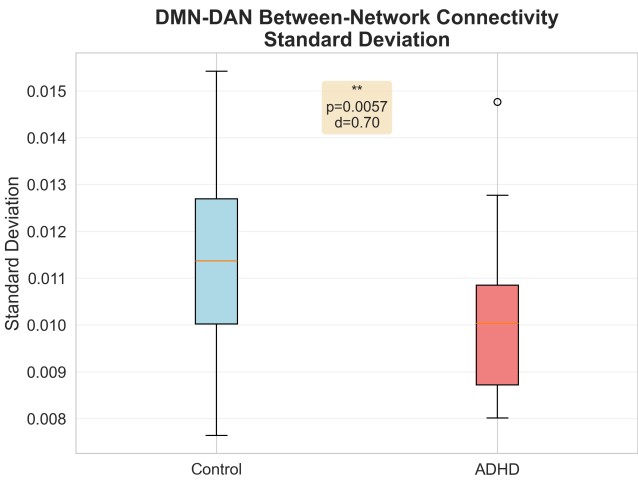

*Figure 16.* Inter-network dynamic connection differences between ADHD and control groups.

**(2) Abnormal dynamic coordination between the Default Mode Network and the Dorsal Attention Network (DMN-DAN).** As shown in Figure 16, the temporal standard deviation of connections between these two networks exhibited a significant difference between groups ($p = 0.0057$, $d = 0.700$), also representing a medium-to-large effect size. The DMN primarily engages in self-referential thinking and internally directed cognition, while the DAN is responsible for externally directed goal attention control. Under normal circumstances, the two networks exhibit a dynamic mutual inhibitory relationship to support flexible switching of cognitive states between introspection and external tasks. The significant alteration in the dynamic amplitude of connections between the DMN and DAN suggests that ADHD patients experience a temporal coordination disorder between these two core systems, which may manifest as insufficient suppression of the default network when external attention is required, leading to task-unrelated thought interference, or abnormal maintenance of the attention system when introspection should be relaxed. This inter-network disorder provides direct dynamic network evidence for the widely observed mind-wandering phenomenon in ADHD. More importantly, this result links the instability

within a single network (DAN) with the abnormal interaction across networks (DMN-DAN coordination), indicating that attention deficits are not a single-layer abnormality but a comprehensive manifestation of multi-level dynamic regulatory disorders.

**(3) Significant decrease in the dynamic range of the prefrontal-parietal pathway** ($p = 0.0015$, $d = 0.779$). As shown in Figure 17, the dynamic range of this pathway in the control group is 0.0419, while in the ADHD group it is 0.0356, representing a decrease of approximately 15.1%. The frontoparietal pathway is a crucial neural basis for executive control and attention regulation. The prefrontal cortex (especially the dorsolateral prefrontal cortex) is responsible for cognitive control and working memory, while the parietal cortex (especially the intraparietal sulcus) is involved in attention orientation and resource allocation. The reduction in the dynamic range of this pathway indicates that the amplitude of changes in connection strength over time is limited, presenting a rigid dynamic characteristic. Further analysis reveals that the standard deviation ($p = 0.0057$, $d = 0.659$) and the coefficient of variation ($p = 0.0082$, $d = 0.363$) of this pathway are also significantly reduced, consistently pointing to a decrease in dynamic flexibility from both absolute and relative dimensions. This result is highly consistent with the theoretical expectation of impaired executive function and attention regulation flexibility in ADHD.

Overall, these six significant metrics exhibit a consistent pattern of limited dynamic flexibility. Whether it is within the dorsal attention network, between the default network and the dorsal attention network, or in the prefrontal-parietal executive control pathway, the ADHD group shows significantly lower temporal dynamic fluctuations across all these key functional connections compared to the control group. This widespread reduction in dynamic range across networks suggests that the brain functional connectivity in ADHD patients lacks the necessary flexibility regulation in the temporal dimension, exhibiting a rigid dynamic characteristic.

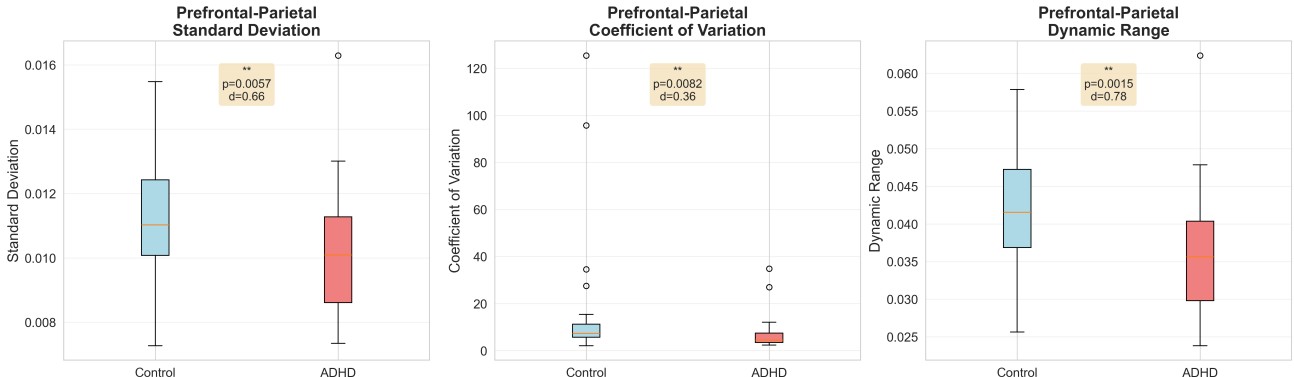

*Figure 17.* Dynamic differences in core functional pathways between ADHD and control groups.

## B.3. Conclusion

This section is based on the graph sequence data from the OHSU subset (valid samples: 45 in the control group and 26 in the ADHD group). Through refined dynamic functional connectivity analysis, we systematically identify the abnormal temporal dynamic characteristics in the brain networks related to attention and executive control in ADHD patients. Statistical tests on 31 dynamic metrics show that 6 metrics reach significance (significance rate 19.4%), which is significantly higher than the expected number of significant metrics under random conditions at the 5% significance level (approximately 1.55). This indicates that the dynamic graph sequences indeed contain stable and non-random disease-related information.

These significant metrics are mainly concentrated in the DAN, the inter-network connections between the DMN and the DAN, and the prefrontal-parietal pathway, forming a multi-level dynamic abnormality pattern centered on the attention control system. Notably, all six significant metrics exhibit a consistent pattern of limited dynamic flexibility: the temporal variability of the DAN is significantly reduced, the dynamic coordination amplitude between the DMN and the DAN is weakened, and the dynamic range, standard deviation, and coefficient of variation of the prefrontal-parietal pathway are systematically decreased. This universal reduction in dynamic range across multiple attention- and executive control-related networks suggests that ADHD patients lack sufficient neural dynamic reserves when maintaining and regulating different cognitive states, making it difficult for them to flexibly adjust network configurations according to task demands.

From an overall structural perspective, the significant results consistently show limited dynamic flexibility: both the

attention-related network (the DAN and its interaction with the DMN) and the executive-attention integration pathway (the prefrontal-parietal pathway) show a significant reduction in dynamic fluctuation amplitude, indicating rigidity of functional connectivity in the temporal dimension. This result is highly consistent with the previous view that ADHD is not a single-pathway disorder but involves dysfunction in multiple large-scale brain systems (Castellanos & Proal, 2012). It also provides new dynamic evidence for understanding the core symptoms of ADHD from the network and system levels.

In terms of functional and clinical significance, the above dynamic abnormalities have a clear correspondence with the core symptom dimensions of ADHD. The limited dynamic flexibility within and between networks provides a direct neurodynamic explanation for persistent attention difficulties and susceptibility to distraction. The reduced dynamic flexibility of the prefrontal-parietal pathway may limit the individual's ability to dynamically adjust attention resources and cognitive strategies according to task demands, thereby exacerbating executive function impairments. This indicates that the dynamic graph features identified in this study not only have good inter-group discriminative ability in statistical terms but also are highly consistent with the behavioral phenotypes of ADHD at the functional level.

From a methodological perspective, the results in this section highlight the unique advantages of dynamic graph sequence analysis. This study demonstrates that the connectivity fluctuation characteristics in the temporal dimension carry important disease-related information. Dynamic graph sequences can reveal how functional connectivity evolves over time rather than merely how strong the connectivity is, providing complementary discriminant clues from the perspective of temporal dynamics. This finding directly supports the necessity of modeling brain functional connectivity as a temporal graph rather than a static graph.

Combining the validation results of strong temporal dependency in Appendix A and the effective differentiation between ADHD and control groups by dynamic graph metrics in this section, this study demonstrates at both the neuroscientific and methodological levels that temporal information across short-term windows not only objectively exists but also has clear clinical discriminative value. This lays a theoretical foundation for the temporal modeling approach based on short-term windows adopted in our proposed SLT-BFGN model.

Figures 18 and 19 illustrate the comparison between short-term brain graph sequences and the adjacent frame time series method (EBIGNN). The adjacent frame method primarily relies on node changes for disease detection, as it cannot directly compute correlations between adjacent frames. Our brain graph sequences, constructed based on short-term windows, characterize short-term neural activity changes in the brain through these windows, thereby yielding rich edge features and community features.

## C. Methodological Details of the SLT-BFGN Model

### C.1. Construction of Brain Map Sequences Based on Short-Term Windows

To characterize short-term functional reorganization and transient activity in the brain, we construct short-term functional window sequences as shown in Figure 3.

Raw fMRI data are parcellated into $N$ ROIs using the AAL atlas, followed by standard preprocessing including detrending, bandpass filtering, and standardization. We divide the preprocessed ROI time-series matrix $X \in \mathbb{R}^{T \times N}$ into $M = \lfloor T/w \rfloor$ non-overlapping windows of length $w$ (typically $w = 10$ time points). The $m$-th window is $X^{(m)} = X[s_m : s_m + w, :]$ with $s_m = (m - 1) \cdot w$.

**Robust Precision Estimation.** For each window $X^{(m)}$, we estimate the precision matrix $\Theta^{(m)}$ using an adaptive strategy that combines GraphicalLassoCV with 3-fold cross-validation and Ledoit-Wolf shrinkage regularization ($\lambda = 10^{-3}$). The estimation procedure automatically adapts to ensure numerical stability and sparsity. Numerical safeguards include imputation of missing values and regularization of near-zero variances.

**Partial Correlation as Weighted Edges.** We convert $\Theta^{(m)}$ into a window-level partial correlation matrix $P^{(m)}$ (diagonal set to 0) as

$$P_{ij}^{(m)} = -\frac{\Theta_{ij}^{(m)}}{\sqrt{\left|\Theta_{ii}^{(m)}\right| \left|\Theta_{jj}^{(m)}\right| + \varepsilon}}, \quad i \neq j, \tag{32}$$

where $\varepsilon = 10^{-12}$ prevents division by zero. The negative sign ensures positive partial correlations correspond to positive edge weights, directly matching our edge weight definition.

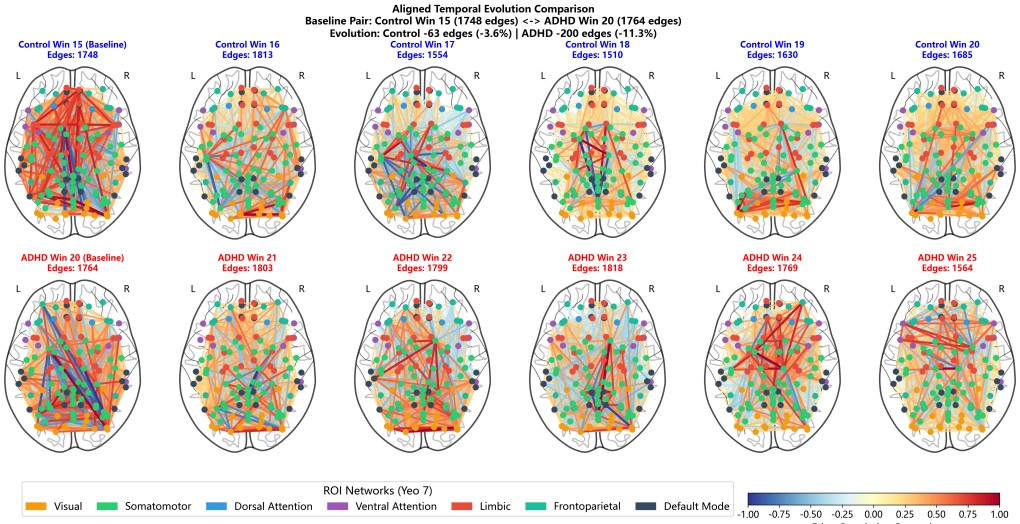

*Figure 18.* Temporal evolution of functional connectivity alignment between ADHD and healthy control samples. Twelve slice images display six consecutive time windows for both groups, with control windows (blue labels) and ADHD windows (red labels) aligned to matching baseline connectivity patterns (identified as control Window 15 and ADHD Window 20 based on Frobenius norm similarity). Each brain connectome is presented in an axial view, displaying regions of interest (ROIs) from the AAL-116 atlas. Node colors represent Yeo 7 functional network divisions, while edge colors denote correlation strength following the *RdYlBu_r* colormap: warm tones (red-yellow) indicate positive correlations and cool tones (blue) indicate negative correlations. Edge thickness reflects connection strength exceeding an automatic threshold. The figures reveal markedly distinct temporal evolution trajectories between the two groups.

**Cross-Window Consistent Sparsification.** A key challenge in dynamic connectivity analysis is ensuring comparability across windows. If each window is thresholded independently, resulting graphs may have drastically different densities, making it difficult to distinguish true temporal changes from thresholding artifacts. We pool absolute upper-triangular values across all windows to form a global distribution:

$$\mathcal{E} = \{|P_{ij}^{(m)}| : \ 1 \leq m \leq M, \ 1 \leq i < j \leq N\}, \tag{33}$$

and compute a global percentile threshold:

$$\tau = \text{percentile}_p(\mathcal{E}),$$
$$A_{ij}^{(m)} = \begin{cases} P_{ij}^{(m)}, & i \neq j \text{ and } |P_{ij}^{(m)}| \geq \tau, \\ 0, & \text{otherwise.} \end{cases} \tag{34}$$

We then enforce symmetry by $\mathbf{A}^{(m)} \leftarrow (\mathbf{A}^{(m)} + (\mathbf{A}^{(m)})^T)/2$. The percentile threshold $\tau = \text{percentile}_p(\mathcal{E})$ controls sparsity. For example, $p = 30$ retains edges with partial correlations in the top 70 percentile. Using global $\tau$ ensures: (1) comparable sparsity across windows; (2) temporal changes reflect genuine differences in partial correlation magnitudes; (3) the graph sequence is suitable for temporal modeling without density confounds.

**Community Sequence.** For each window graph $A^{(m)}$, we perform community detection using the Louvain algorithm (greedy modularity maximization) with edge weights from the adjacency matrix, optimizing:

$$Q = \frac{1}{2W} \sum_{ij} \left[ A_{ij} - \frac{k_i k_j}{2W} \right] \delta(c_i, c_j), \tag{35}$$

where $W = \sum_{ij} A_{ij}$, $k_i = \sum_j A_{ij}$, and $\delta(c_i, c_j) = 1$ if $c_i = c_j$. This yields community labels $c^{(m)} \in \mathbb{Z}^N$ (isolated nodes assigned $-1$). The final output includes the time-indexed graph sequence $\{A^{(m)}\}_{m=1}^M$, the community sequence $\{c^{(m)}\}_{m=1}^M$, and window start indices.

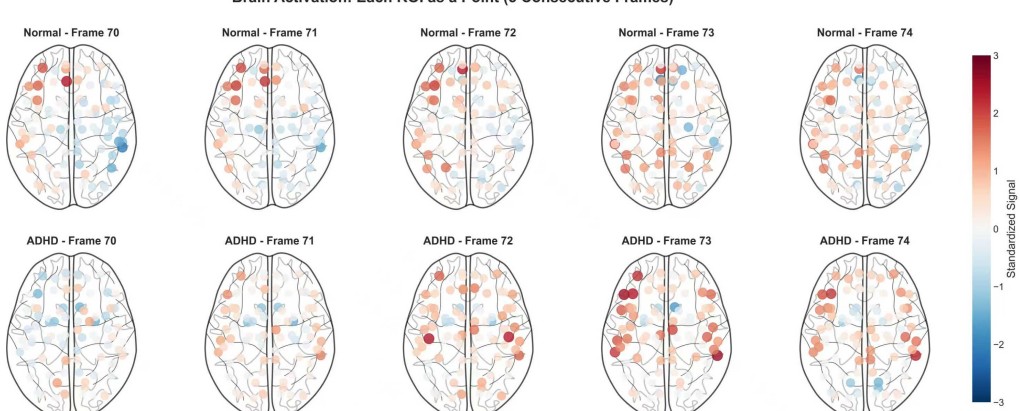

*Figure 19.* Brain maps comparing ADHD and healthy control samples using the adjacent frame method (EBIGNN). EBIGNN primarily relies on node changes for disease detection since it cannot directly calculate correlations between adjacent frames.

### C.2. Short-Term State and Temporal Dependency Encoder

To characterize short-term sequence features of brain function, we design a hierarchical encoder that first encodes each time window into a compact state representation, then models temporal dependencies using Transformer architecture.

#### C.2.1. SHORT-TERM STATE ENCODER

For each time window $m$, we construct node-level features by combining degree statistics, spatial information, and topological clustering.

**Degree Dynamics and Neighborhood Geometry.** For node $i$ in window $m$, we compute:

$$
\begin{aligned}
deg_i^{(m)} &= \sum_{j=1}^{N} \left| A_{ij}^{(m)} \right|, \\
\Delta deg_i^{(m)} &= deg_i^{(m)} - deg_i^{(m-1)}.
\end{aligned}
\tag{36}
$$

We also compute the neighborhood geometric center:

$$
neigh\_coord_i^{(m)} = \sum_{j=1}^{N} \frac{A_{ij}^{(m)}}{\max(deg_i^{(m)}, 1)} \, coord_j,
\tag{37}
$$

where $coord_j \in \mathbb{R}^3$ is the 3D MNI coordinate of ROI $j$. This feature computes a weighted average of the spatial locations of node $i$'s neighbors. If node $i$ is primarily connected to frontal regions, $neigh\_coord_i^{(m)}$ will be located anteriorly; if connections span distant regions, the center reflects this spatial diversity. The denominator $\max(deg_i^{(m)}, 1)$ prevents division by zero for isolated nodes. For windows where previous degree is unavailable, $\Delta deg_i$ is set to zero.

**Feature Aggregation.** The raw node feature vector is $f_i^{(m)} = [deg_i^{(m)}, \, coord_i, \, neigh\_coord_i^{(m)}, \, emb(c_i^{(m)}), \, \Delta deg_i^{(m)}]$, where $emb(c_i^{(m)}) \in \mathbb{R}^{d_{\mathrm{emb}}}$ is a learnable embedding mapping community labels to dense vectors. For nodes with label $-1$, we use a dedicated "null community" embedding. These features are transformed through:

1. Layer normalization: $\widetilde{f}_i^{(m)} = \mathrm{LayerNorm}(f_i^{(m)})$

2. Linear projection: $h_i^{(m)} = W_1 \widetilde{f}_i^{(m)} + b_1 \in \mathbb{R}^{d_{\mathrm{hidden}}}$

3. Residual feedforward network:

$$
\begin{aligned}
h_i'^{(m)} &= \mathrm{ReLU}(W_2 h_i^{(m)} + b_2), \\
x_i^{(m)} &= \mathrm{LayerNorm}(h_i^{(m)} + W_3 h_i'^{(m)} + b_3) \in \mathbb{R}^d.
\end{aligned}
\tag{38}
$$

Average pooling yields window-level embedding $z^{(m)} = \frac{1}{N} \sum_{i=1}^{N} x_i^{(m)} \in \mathbb{R}^d$. This serves as a compact, permutation-invariant representation of the brain state in window $m$.

**Community-Based Anomaly Scores.** To enhance detection of subtle organizational differences, we introduce anomaly scores highlighting windows with rare community structures. The intuition is that certain community patterns may be uncommon but highly discriminative for ADHD (e.g., atypical segregation of attention and default-mode networks). We estimate empirical frequency $\pi(c)$ of each community label $c$ by counting occurrences across all training windows. For window $m$, the anomaly score is:

$$s^{(m)} = -\frac{1}{|\mathcal{V}^{(m)}|} \sum_{i \in \mathcal{V}^{(m)}} \log \pi(c_i^{(m)}), \tag{39}$$

where $\mathcal{V}^{(m)} = \{i \mid c_i^{(m)} \neq -1\}$. High scores indicate rare communities. The model can use these scores (concatenated with other features) to upweight potentially discriminative windows.

### C.2.2. TEMPORAL DEPENDENCY ENCODER

**Positional Encoder.** The window embeddings $\{z^{(m)}\}_{m=1}^{M}$ are order-agnostic. To explicitly encode temporal order, we inject sinusoidal positional encoding:

$$\begin{aligned}
\text{PE}_{2i}^{(m)} &= \sin\left(\frac{m}{10000^{2i/d}}\right), \\
\text{PE}_{2i+1}^{(m)} &= \cos\left(\frac{m}{10000^{2i/d}}\right),
\end{aligned} \tag{40}$$

where $m \in \{1, 2, \ldots, M\}$ is the window index and $i \in \{0, 1, \ldots, \lfloor d/2 \rfloor - 1\}$ indexes dimension pairs. The wavelength in dimension $2i$ is $2\pi \cdot 10000^{2i/d}$, varying exponentially from $2\pi$ to $2\pi \cdot 10000$, allowing the model to easily attend to relative positions. The positionally-encoded representation is:

$$z'^{(m)} = z^{(m)} + \text{PE}^{(m)}. \tag{41}$$

Addition (rather than concatenation) keeps dimensionality fixed and allows position and content information to interact throughout all layers.

**Classification Token Construction.** A learnable classification token $z_{\text{cls}} \in \mathbb{R}^d$ (randomly initialized, updated via backpropagation) serves as a dedicated aggregator that can attend to all positions and accumulate classification-relevant information. It is prepended to form:

$$S = [z_{\text{cls}}; z'^{(1)}; z'^{(2)}; \ldots; z'^{(M)}] \in \mathbb{R}^{(M+1) \times d}. \tag{42}$$

For variable-length sequences, we pad to maximum batch length $M_{\max}$ with zeros and construct a boolean mask marking valid positions, preventing attention to padded positions.

**Transformer Encoding.** $L$ stacked Transformer layers process $S$:

$$\begin{aligned}
\widetilde{S}^{(\ell)} &= \text{LayerNorm}(S^{(\ell-1)} + \text{MultiHeadAttention}(S^{(\ell-1)})), \\
S^{(\ell)} &= \text{LayerNorm}(\widetilde{S}^{(\ell)} + \text{FFN}(\widetilde{S}^{(\ell)})),
\end{aligned} \tag{43}$$

where $S^{(0)} = S$. The multi-head attention computes, for $H$ heads:

$$\text{head}_h = \text{softmax}\left(\frac{Q_h K_h^\top}{\sqrt{d_k}} + \text{mask}\right) V_h, \tag{44}$$

where $d_k = d/H$ and mask sets invalid positions to $-\infty$ before softmax. Outputs are concatenated and projected: $\text{MultiHeadAttention} = \text{Concat}(\text{head}_1, \ldots, \text{head}_H) W_O$. Shallow layers capture local temporal relationships, while deeper layers extract long-range dependencies and global trends.

**Temporal Aggregation.** The classification token output aggregates information from all windows:

$$y_{\text{cls}} = S^{(L)}[0, :] \in \mathbb{R}^d. \tag{45}$$

Through attention across all layers, $y_{\text{cls}}$ has accumulated information from all $M$ windows. The aggregation is learned rather than fixed (unlike simple averaging), with the model learning to attend more to discriminative windows.

### C.3. Brain Function Memory

To capture associations between transient brain activity and historical patterns across patients, we introduce learnable memory modules in both branches. The key insight is that prototypical brain patterns (e.g., certain temporal dynamics or structural configurations) recur across multiple patients and are informative for diagnosis.

Each memory maintains a learnable matrix $\mathcal{M} \in \mathbb{R}^{K \times d_m}$ with $K$ memory slots storing $d_m$-dimensional patterns. The matrix is randomly initialized and updated during training via gated write, but fixed at test time (used only for reading). At test time, the memory serves as a reference bank of prototypical patterns learned from training data. The attention mechanism allows test samples to flexibly retrieve relevant patterns, providing robust generalization to unseen subjects.

**Memory Reading.** Given input $x \in \mathbb{R}^d$, we query the memory:

$$
\begin{aligned}
q &= W_{\text{proj}} \cdot \text{LayerNorm}(x) \in \mathbb{R}^{d_m}, \\
\boldsymbol{\alpha} &= \text{softmax}(q\mathcal{M}^{\top}) \in \mathbb{R}^{K}, \\
r &= \boldsymbol{\alpha}\mathcal{M} \in \mathbb{R}^{d_m},
\end{aligned}
\tag{46}
$$

where $\alpha_k$ represents similarity between the current state and the $k$-th memory slot. High attention weight indicates the current state resembles the stored pattern. The readout $r$ injects long-term contextual information into the current representation.

**Memory Writing.** During training, we update memory via gated batch-averaged signals to prevent instability and catastrophic forgetting. For each sample, we compute write-in signal $w_k = W_{\text{write}} \cdot [q; \mathcal{M}_{k,:}] + b_{\text{write}}$ and gate $g_k = \sigma(W_{\text{gate}} \cdot [q; \mathcal{M}_{k,:}] + b_{\text{gate}})$ for each slot. To stabilize training, we accumulate over mini-batch of $B$ samples:

$$
\begin{aligned}
\overline{w}_k &= \frac{1}{B} \sum_{b=1}^{B} \alpha_k^{(b)} w_k^{(b)}, \\
\overline{g}_k &= \frac{1}{B} \sum_{b=1}^{B} \alpha_k^{(b)} g_k^{(b)},
\end{aligned}
\tag{47}
$$

where attention-weighting ensures slots are primarily updated by samples that strongly attend to them, helping to specialize slots for different pattern types. The memory update is:

$$
\mathcal{M} \leftarrow (1 - \overline{G}) \odot \mathcal{M} + \overline{G} \odot \overline{W},
\tag{48}
$$

where $\overline{G}, \overline{W} \in \mathbb{R}^{K \times d_m}$ are batch-averaged gates and write-ins. Over training, slots specialize to different prototypical patterns (e.g., ADHD vs. healthy vs. subtypes).

In the short-term branch, memory operates on $y_{\text{cls}}$ ($x = y_{\text{cls}}$). Since $y_{\text{cls}}$ encodes temporal dynamics from positionally-encoded windows, memory slots learn to store prototypes of recurring temporal brain dynamics patterns. For example, a slot might capture patterns where frontal connectivity increases while posterior connectivity decreases over time, associated with attentional lapses in ADHD.

### C.4. GNN-based Long-Term Brain Function Feature Extraction Network

The long-term branch extracts features from a stable, time-averaged connectivity graph summarizing overall functional architecture.

**Global Graph Construction.** The global adjacency $A \in \mathbb{R}^{N \times N}$ is constructed by applying the same robust precision estimation and percentile-based thresholding to the entire time series (rather than windows). Since $T \gg N$ for full-session data, we use GraphicalLassoCV with the same safeguards. We convert precision to global partial correlation matrix $P$, apply the same percentile thresholding to obtain sparse $A$, and perform community detection to obtain global labels $c \in \mathbb{Z}^N$. This global graph captures stable long-term connectivity that is robust to transient fluctuations.

**Node Feature Initialization.** Each node $i$ is initialized with: connectivity profile $A_{i,:} \in \mathbb{R}^N$ (encoding functional connectivity pattern), degree $d_i = \sum_{j=1}^{N} |A_{ij}|$ (overall connectivity strength), spatial coordinates $\text{coord}_i \in \mathbb{R}^3$, and community embedding $\text{emb}(c_i) \in \mathbb{R}^{d_{\text{emb}}}$ (distinct from short-term embeddings, allowing different representations for static vs. dynamic roles). These are concatenated and gated with community information:

$$
h_i^{(0)} = f_i + \sigma(W_{\text{gate}}^{\text{comm}} f_i) \odot \text{emb}(c_i),
\tag{49}
$$

where the gate determines how much community embedding is mixed in. The result is then projected to hidden dimension $d$.

**GCN Layers with Residual Connections.** We use GCN layers for hierarchical message passing. To ensure stability, we add self-loops and construct symmetric normalized adjacency:

$$\widehat{A} = D^{-1/2}(A+I)D^{-1/2}, \qquad D_{ii} = \sum_{j=1}^{N} |(A+I)_{ij}|. \tag{50}$$

The normalization ensures aggregated messages have consistent scale, preventing instability when stacking layers. For layer $\ell$ (typically $L_{\text{GNN}} = 3$ to $5$):

$$\begin{aligned}
\widetilde{H}^{(\ell)} &= \widehat{A}H^{(\ell-1)}W_{\text{GCN}}^{(\ell)}, \\
H^{(\ell)} &= \text{ReLU}(\text{LayerNorm}(\widetilde{H}^{(\ell)} + H^{(\ell-1)})),
\end{aligned} \tag{51}$$

where $\widehat{A}H^{(\ell-1)}$ performs weighted neighbor aggregation, followed by linear transformation, layer normalization, activation, and residual connection. Residual connections mitigate over-smoothing and facilitate gradient flow.

**Graph-Level Pooling.** To obtain graph-level embedding, we combine three pooling strategies:

- Mean pooling: $g_{\text{mean}} = \frac{1}{N}\sum_{i=1}^{N} h_i^{(L_{\text{GNN}})}$ (smooth, robust to outliers)

- Max pooling: $g_{\text{max}}[j] = \max_{i=1}^{N} h_i^{(L_{\text{GNN}})}[j]$ (captures salient features)

- Attention pooling: $g_{\text{attn}} = \sum_{i=1}^{N} \beta_i h_i^{(L_{\text{GNN}})}$ where $\beta_i = \frac{\exp(e_i)}{\sum_j \exp(e_j)}$ and $e_i = v_{\text{attn}}^{\top}\tanh(W_{\text{attn}}h_i^{(L_{\text{GNN}})} + b_{\text{attn}})$ (learned, task-specific weighting)

These are concatenated as $g_{\text{combined}} = [g_{\text{mean}}; g_{\text{max}}; g_{\text{attn}}] \in \mathbb{R}^{3d}$ and projected: $g = W_{\text{pool}}g_{\text{combined}} + b_{\text{pool}} \in \mathbb{R}^d$.

In the long-term branch, memory operates on graph embedding $g$ ($x = g$). Memory slots learn prototypes of recurring static topological patterns, e.g., strong fronto-parietal connectivity combined with weak default-mode connectivity that is characteristic of certain ADHD subgroups.

## C.5. Feature Fusion

The fusion module combines complementary information from both branches through decision-level fusion, preserving branch independence during feature learning while leveraging both at decision stage.

**Branch-Specific Classification Heads.** Short-term branch: Transformer's CLS output $y^L$, brain function memory readout $r^L$, and projected graph statistics $p_L$ (including mean/std/max of window-level degrees, mean/std of community anomaly scores, and total number of windows) are concatenated as $h^L = [y^L; r^L; p_L]$ and mapped to a score: $s^L = w^{L\top}h^L + b^L$.

Long-term branch: GCN embedding $g$, structural memory readout $r^G$, and projected global statistics $p^G$ (including global degree statistics, clustering coefficients, community modularity) are concatenated as $h^G = [g; r^G; p^G]$ and mapped to a score: $s^G = w^{G\top}h^G + b^G$.

**Fusion Strategies.** We support: (1) Simple averaging $s_{fus} = (s^L + s^G)/2$ (simplest, most robust); (2) Weighted linear fusion with softmax-normalized learnable weights $w^L, w^G$; (3) Nonlinear fusion via feedforward network. Empirically, simple averaging and weighted linear fusion perform comparably well, while nonlinear fusion tends to overfit on smaller datasets, so we primarily use simple averaging.

**Dual-Branch Training with Staged Weight Adjustment.** To prevent branch collapse (one branch dominating while the other is ignored), we train with dual-branch loss:

$$\mathcal{L} = w^L \mathcal{L}_{=task}(s^L, y) + w^G \mathcal{L}_{=task}(s^G, y), \tag{52}$$

where loss weights $w^L, w^G$ evolve through three stages:

1. **Warmup phase (epochs 1-10)**: With fixed equal weights $w^L = w^G = 0.5$, both branches develop basic representations without one dominating during early unstable training.

2. **Learning phase (epochs 11-50)**: Weights are treated as learnable parameters (initialized at 0.5) and optimized with regularization $\mathcal{L}_{\text{reg}} = \lambda_{\text{reg}}(w^L - w^G)^2$ (e.g., $\lambda_{\text{reg}} = 0.01$) encouraging balance. Weights are constrained to satisfy $w^L + w^G = 1$, $w^L, w^G \geq 0$ via softmax reparameterization. The model learns to dynamically adjust weights based on relative difficulty and informativeness of each branch.

3. **Stable phase (epochs 51 onwards)**: Weights are frozen at learned values, preventing drift during final fine-tuning and ensuring stable convergence.

## D. Pseudocode and Notation

This appendix provides detailed pseudocode for all components of our ADHD detection model.

### D.1. Construction of Brain Map Sequences Based on Short-Term Windows

---

**Algorithm 1** Short-Term Window-Based Graph Sequence Construction

---

**Input** : fMRI_data: 4D fMRI volume; atlas: parcellation; $w$: window size; $\theta$: sparsity percentile
**Output**: $\mathbf{A}_{\text{ST}} \in \mathbb{R}^{M \times N \times N}$: temporal connectivity; $\mathbf{C}_{\text{ST}} \in \mathbb{Z}^{M \times N}$: community sequence
$\mathbf{X} \leftarrow$ ExtractTimeSeries(fMRI_data, atlas)  $\qquad\qquad (T_{\text{total}}, N)$
 $M \leftarrow \lfloor T_{\text{total}}/w \rfloor$;    starts $\leftarrow [0, w, 2w, \ldots, (M-1) \cdot w]$
 **// Compute partial correlations for each window**
 all_values $\leftarrow []$
 **for** $m \leftarrow 1$ **to** $M$ **do**
   $\mathbf{X}^{(m)} \leftarrow \mathbf{X}[\text{starts}[m-1] : \text{starts}[m-1] + w, :]$
   $\mathbf{P}^{(m)} \leftarrow$ ComputePartialCorrelation($\mathbf{X}^{(m)}$)
   all_values.extend($\{|\mathbf{P}^{(m)}[i,j]| : i < j\}$)
**// Global thresholding**
 $\tau \leftarrow$ Percentile(all_values, $\theta$)
 **for** $m \leftarrow 1$ **to** $M$ **do**
   $\mathbf{A}_{\text{ST}}[m] \leftarrow$ Sparsify($\mathbf{P}^{(m)}, \tau$)
   $\mathbf{C}_{\text{ST}}[m] \leftarrow$ GreedyModularity(BuildGraph($\mathbf{A}_{\text{ST}}[m]$))
**return** $\mathbf{A}_{\text{ST}}, \mathbf{C}_{\text{ST}}$

---

### D.2. Short-Term Branch Components

---

**Algorithm 2** Short-Term State Encoder (Window Encoder)

---

**Input** : $\mathbf{A}_{\text{curr}}^{(m)}, \mathbf{A}_{\text{prev}}^{(m-1)} \in \mathbb{R}^{B \times N \times N}$: adjacency; $\mathbf{X}_{\text{coords}} \in \mathbb{R}^{B \times N \times 3}$; $\mathbf{c}^{(m)} \in \mathbb{Z}^{B \times N}$: communities
**Output**: $\mathbf{z}^{(m)} \in \mathbb{R}^{B \times d}$: window-level representation
**// Degree and temporal features**
 $\mathbf{deg}^{(m)} \leftarrow \sum_j |\mathbf{A}_{\text{curr}}^{(m)}[:,:,j]|$; $\mathbf{deg}^{(m-1)} \leftarrow \sum_j |\mathbf{A}_{\text{prev}}^{(m-1)}[:,:,j]|$
 $\Delta \mathbf{deg}^{(m)} \leftarrow \mathbf{deg}^{(m)} - \mathbf{deg}^{(m-1)}$
 **// Neighborhood aggregation**
 $\hat{\mathbf{A}} \leftarrow \mathbf{A}_{\text{curr}}^{(m)}/(\mathbf{deg}^{(m)}.\text{unsqueeze}(-1) + \epsilon)$
 $\mathbf{neigh\_coord}^{(m)} \leftarrow \hat{\mathbf{A}} \cdot \mathbf{X}_{\text{coords}}$
 **// Feature transformation**
 $\mathbf{E}_{\text{comm}} \leftarrow$ Embedding($\mathbf{c}^{(m)} + 1$)
 $\mathbf{f}^{(m)} \leftarrow [\mathbf{deg}^{(m)} \| \mathbf{X}_{\text{coords}} \| \mathbf{neigh\_coord}^{(m)} \| \mathbf{E}_{\text{comm}} \| \Delta \mathbf{deg}^{(m)}]$
 $\mathbf{x}^{(m)} \leftarrow$ Linear(LayerNorm($\mathbf{f}^{(m)}$))
 $\mathbf{x}_{\text{ffn}}^{(m)} \leftarrow$ Linear$_2$(GELU(Linear$_1$($\mathbf{x}^{(m)}$)))
 $\mathbf{x}_{\text{out}}^{(m)} \leftarrow$ LayerNorm($\mathbf{x}^{(m)} + \mathbf{x}_{\text{ffn}}^{(m)}$)
 $\mathbf{z}^{(m)} \leftarrow$ mean($\mathbf{x}_{\text{out}}^{(m)}$, dim $= 1$)
 **return** $\mathbf{z}^{(m)}$

---

---

**Algorithm 3** Temporal Dependency Encoder – Part I: Window Encoding

---

**Input** : $\mathbf{A}_{\text{ST}} \in \mathbb{R}^{B \times M \times N \times N}$: temporal graphs; $\mathbf{X}_{\text{coords}}$; $\mathbf{C}_{\text{ST}}$; $\mathbf{l} \in \mathbb{Z}^{B}$: valid lengths; $\boldsymbol{\pi}$: community frequency
**Output** : $\{\mathbf{z}^{(m)}\}_{m=1}^{M} \in \mathbb{R}^{B \times M \times d}$: window embeddings; $\mathbf{p}_{\text{ST}} \in \mathbb{R}^{B \times d_s}$: statistics

// **Window-wise encoding**
  $\{\mathbf{z}^{(m)}\}_{m=1}^{M} \leftarrow \mathbf{0}_{B \times M \times d}$
  **for** $m \leftarrow 1$ **to** $M$ **do**
    $\mathbf{A}_{\text{curr}}^{(m)} \leftarrow \mathbf{A}_{\text{ST}}[:, m, :, :]$
    $\mathbf{A}_{\text{prev}}^{(m-1)} \leftarrow \mathbf{A}_{\text{ST}}[:, m-1, :, :]$ if $m > 1$ else $\mathbf{A}_{\text{curr}}^{(m)}$
    $\mathbf{z}^{(m)} \leftarrow \text{ShortTermStateEncoder}(\mathbf{A}_{\text{curr}}^{(m)}, \mathbf{A}_{\text{prev}}^{(m-1)}, \mathbf{X}_{\text{coords}}, \mathbf{C}_{\text{ST}}[:, m, :])$

// **Community anomaly statistics**
  **for** $b \leftarrow 1$ **to** $B$ **do**
    $s^{(b)} \leftarrow 0$
    **for** $m \leftarrow 1$ **to** $\mathbf{l}_b$ **do**
      $\mathcal{V}^{(m)} \leftarrow \{i \mid \mathbf{C}_{\text{ST}}[b, m, i] \neq -1\}$
      $s^{(m)} \leftarrow -\frac{1}{|\mathcal{V}^{(m)}|} \sum_{i \in \mathcal{V}^{(m)}} \log(\boldsymbol{\pi}(\mathbf{C}_{\text{ST}}[b, m, i]) + \epsilon)$
      $s^{(b)} \leftarrow s^{(b)} + s^{(m)}$
    $s^{(b)} \leftarrow s^{(b)} / \mathbf{l}_b$

  $\mathbf{p}_{\text{ST}} \leftarrow \text{Linear}(\mathbf{s})$                                project statistics
  **return** $\{\mathbf{z}^{(m)}\}_{m=1}^{M}, \mathbf{p}_{\text{ST}}$

---

**Algorithm 4** Temporal Dependency Encoder – Part II: Transformer Encoding

---

**Input** : $\{\mathbf{z}^{(m)}\}_{m=1}^{M} \in \mathbb{R}^{B \times M \times d}$: window embeddings; $\mathbf{l} \in \mathbb{Z}^{B}$: valid lengths
**Output** : $\mathbf{y}_{\text{cls}} \in \mathbb{R}^{B \times d}$: temporal-encoded representation

// **Positional encoding**
  $\mathbf{PE}^{(m)} \leftarrow \text{PositionalEncoding}(M, d) \quad \forall m \in [1, M]$
  $\mathbf{z}'^{(m)} \leftarrow \mathbf{z}^{(m)} + \mathbf{PE}^{(m)} \quad \forall m$
// **CLS token construction**
  $\mathbf{z}_{\text{cls}} \leftarrow \text{LearnableParameter}(d)$
  $\mathbf{S} \leftarrow [\mathbf{z}_{\text{cls}}; \mathbf{z}'^{(1)}; \mathbf{z}'^{(2)}; \ldots; \mathbf{z}'^{(M)}]$                         $(B, M+1, d)$
// **Transformer encoding**
  **for** $\ell \leftarrow 1$ **to** $L_{trans}$ **do**
    $\mathbf{Q}, \mathbf{K}, \mathbf{V} \leftarrow \text{Linear}_Q(\mathbf{S}^{(\ell-1)}), \text{Linear}_K(\mathbf{S}^{(\ell-1)}), \text{Linear}_V(\mathbf{S}^{(\ell-1)})$
    $\text{mask} \leftarrow \text{CreatePaddingMask}(\mathbf{l}, M+1)$
    $\boldsymbol{\alpha} \leftarrow \text{Softmax}((\mathbf{Q}\mathbf{K}^{\top} / \sqrt{d_k}) + \text{mask})$
    $\mathbf{S}_{\text{attn}} \leftarrow \text{Concat}(\boldsymbol{\alpha}\mathbf{V}) \cdot \mathbf{W}_O$
    $\mathbf{S}^{(\ell)} \leftarrow \text{LayerNorm}(\mathbf{S}^{(\ell-1)} + \mathbf{S}_{\text{attn}})$
    $\mathbf{S}_{\text{ffn}} \leftarrow \text{Linear}_2(\text{GELU}(\text{Linear}_1(\mathbf{S}^{(\ell)})))$
    $\mathbf{S}^{(\ell)} \leftarrow \text{LayerNorm}(\mathbf{S}^{(\ell)} + \mathbf{S}_{\text{ffn}})$

// **Extract CLS token output**
  $\mathbf{y}_{\text{cls}} \leftarrow \mathbf{S}^{(L_{\text{trans}})}[:, 0, :]$                          temporal-encoded representation
  **return** $\mathbf{y}_{\text{cls}}$

---

**Algorithm 5** Brain Function Memory Module (Short-Term Branch)

---

**Input** : $\mathbf{y}_{\text{cls}} \in \mathbb{R}^{B \times d}$: temporal-encoded representation; $\mathcal{M}_{\text{ST}} \in \mathbb{R}^{K \times d_m}$: memory bank
**Output** : $\mathbf{r}_{\text{ST}} \in \mathbb{R}^{B \times d_m}$: memory-augmented features

// **Memory reading**
  $\mathbf{q} \leftarrow \text{Linear}_{\text{proj}}(\text{LayerNorm}(\mathbf{y}_{\text{cls}}))$                         $(B, d_m)$
  $\boldsymbol{\alpha} \leftarrow \text{Softmax}(\mathbf{q}\mathcal{M}_{\text{ST}}^{\top} / \sqrt{d_m}, \text{dim} = -1)$
  $\mathbf{r}_{\text{mem}} \leftarrow \boldsymbol{\alpha}\mathcal{M}_{\text{ST}}$
  $\mathbf{g} \leftarrow \sigma(\text{Linear}_g([\mathbf{q} \| \mathbf{r}_{\text{mem}}]))$
  $\mathbf{r}_{\text{ST}} \leftarrow \mathbf{g} \odot \mathbf{r}_{\text{mem}} + (1 - \mathbf{g}) \odot \mathbf{q}$
  **return** $\mathbf{r}_{\text{ST}}$

---

---

**Algorithm 6** Short-Term Branch Classifier

---

**Input** : $\mathbf{y}_{\text{cls}} \in \mathbb{R}^{B \times d}$: temporal representation; $\mathbf{r}_{\text{ST}} \in \mathbb{R}^{B \times d_m}$: memory features; $\mathbf{p}_{\text{ST}} \in \mathbb{R}^{B \times d_s}$: statistics
**Output** : $s^{\text{ST}} \in \mathbb{R}^{B \times 2}$: class logits
$\mathbf{h}_{\text{ST}} \leftarrow [\mathbf{y}_{\text{cls}} \| \mathbf{r}_{\text{ST}} \| \mathbf{p}_{\text{ST}}]$
$\quad \mathbf{x} \leftarrow \text{Dropout}(\text{ReLU}(\text{Linear}_1(\mathbf{h}_{\text{ST}})))$
$\quad \mathbf{x} \leftarrow \text{Dropout}(\text{ReLU}(\text{Linear}_2(\mathbf{x})))$
$\quad s^{\text{ST}} \leftarrow \text{Linear}_{\text{out}}(\mathbf{x})$
$\quad$ **return** $s^{\text{ST}}$

---

## D.3. Long-Term Branch Components

---

**Algorithm 7** Long-Term GCN Encoder

---

**Input** : $\mathbf{A}_{\text{LT}} \in \mathbb{R}^{B \times N \times N}$: connectivity; $\mathbf{X}_{\text{coords}} \in \mathbb{R}^{B \times N \times 3}$: coordinates; $\mathbf{C} \in \mathbb{Z}^{B \times N}$: communities; $L$: layers
**Output** : $\mathbf{H}_{\text{LT}} \in \mathbb{R}^{B \times N \times d_h}$: node representations
// **Initial feature construction**
$\quad$ **for** $b \leftarrow 1$ **to** $B$ **do**
$\qquad \mathbf{d}_b \leftarrow \sum_j \mathbf{A}_{\text{LT}}[b, :, j]$ $\hfill$ degree, $(N,)$
$\qquad \mathbf{X}_{\text{init}}[b] \leftarrow [\mathbf{A}_{\text{LT}}[b] \| \mathbf{d}_b \| \mathbf{X}_{\text{coords}}[b]]$

$\mathbf{H}^{(0)} \leftarrow \text{LayerNorm}(\text{Linear}(\mathbf{X}_{\text{init}}))$ $\hfill (B, N, d_h)$
$\quad$ // **Community-enhanced features**
$\quad \mathbf{E}_{\text{comm}} \leftarrow \text{CommunityEmbed}(\mathbf{C})$
$\quad \mathbf{g} \leftarrow \sigma(\text{Linear}([\mathbf{H}^{(0)} \| \mathbf{E}_{\text{comm}}]))$
$\quad \mathbf{H}^{(0)} \leftarrow \mathbf{H}^{(0)} + \mathbf{g} \odot \mathbf{E}_{\text{comm}}$
$\quad$ // **Multi-layer GCN**
$\quad$ **for** $\ell \leftarrow 1$ **to** $L$ **do**
$\qquad \tilde{\mathbf{A}} \leftarrow \mathbf{A}_{\text{LT}} + \mathbf{I}_N; \quad \tilde{\mathbf{D}} \leftarrow \text{Diag}(\sum_j \tilde{\mathbf{A}}_{:,j})$
$\qquad \hat{\mathbf{A}} \leftarrow \tilde{\mathbf{D}}^{-1/2} \tilde{\mathbf{A}} \tilde{\mathbf{D}}^{-1/2}$
$\qquad \mathbf{H}^{(\ell)} \leftarrow \sigma(\hat{\mathbf{A}} \mathbf{H}^{(\ell-1)} \mathbf{W}^{(\ell)})$
$\qquad \mathbf{H}^{(\ell)} \leftarrow \text{Dropout}(\text{LayerNorm}(\mathbf{H}^{(\ell)} + \mathbf{H}^{(\ell-1)}))$

**return** $\mathbf{H}_{\text{LT}} \leftarrow \mathbf{H}^{(L)}$

---

**Algorithm 8** Static Long-Term Graph Construction

---

**Input** : fMRI_data: 4D fMRI volume; atlas: brain parcellation; $\theta$: sparsity percentile
**Output** : $\mathbf{A}_{\text{LT}} \in \mathbb{R}^{N \times N}$: static connectivity; $\mathbf{C}_{\text{LT}} \in \mathbb{Z}^N$: community labels; coords $\in \mathbb{R}^{N \times 3}$
// **Step 1: Time-series extraction**
$\quad$ masker $\leftarrow$ NiftiLabelsMasker(atlas, standardize=True)
$\quad \mathbf{X} \leftarrow$ masker.fit_transform(fMRI_data) $\hfill (T, N)$
$\quad$ // **Step 2: Precision matrix estimation**
$\quad$ **if** $T > N$ **then**
$\qquad \Theta \leftarrow \text{GraphicalLassoCV}(\mathbf{X}).\text{precision\_}$
$\quad$ **else**
$\qquad \Sigma \leftarrow \text{LedoitWolf}(\mathbf{X}).\text{covariance\_} + 10^{-3} \mathbf{I}_N$
$\qquad \Theta \leftarrow \text{PseudoInverse}(\Sigma)$

$\mathbf{D} \leftarrow \text{Diag}(\sqrt{|\text{diag}(\Theta)|} + 10^{-12})$
$\quad \mathbf{P} \leftarrow -\Theta/(\mathbf{D} \cdot \mathbf{D}^\top); \quad \mathbf{P} \leftarrow \text{SetDiag}(\mathbf{P}, 0)$
$\quad$ // **Step 3: Sparsification**
$\quad \tau \leftarrow \text{Percentile}(\{|\mathbf{P}_{ij}| : i < j\}, \theta)$
$\quad \mathbf{A}_{\text{LT}}[i, j] \leftarrow \mathbf{P}_{ij} \cdot \mathbb{I}[|\mathbf{P}_{ij}| \geq \tau] \quad \forall i, j$
$\quad \mathbf{A}_{\text{LT}} \leftarrow (\mathbf{A}_{\text{LT}} + \mathbf{A}_{\text{LT}}^\top)/2$
$\quad$ // **Step 4: Community detection**
$\quad G \leftarrow \text{BuildWeightedGraph}(\mathbf{A}_{\text{LT}})$
$\quad \mathbf{C}_{\text{LT}} \leftarrow \text{GreedyModularity}(G)$
$\quad$ // **Step 5: Extract coordinates**
$\quad$ coords $\leftarrow$ ComputeCentroids(atlas)
$\quad$ **return** $\mathbf{A}_{\text{LT}}, \mathbf{C}_{\text{LT}}$, coords

---

---

**Algorithm 9** Long-Term Memory Module

---

**Input** : $\mathbf{H}_{\mathrm{LT}} \in \mathbb{R}^{B \times N \times d_h}$: node representations; $\mathcal{M}_{\mathrm{LT}} \in \mathbb{R}^{K \times d_m}$: memory bank
**Output** : $\mathbf{r}_{\mathrm{LT}} \in \mathbb{R}^{B \times d_m}$: memory-augmented features
$\mathbf{q} \leftarrow \mathrm{Linear}_Q(\mathrm{MeanPool}(\mathbf{H}_{\mathrm{LT}}, \mathrm{dim} = 1))$ $\hfill (B, d_m)$
$\quad \mathbf{s} \leftarrow \mathbf{q}\mathcal{M}_{\mathrm{LT}}^{\top}/\sqrt{d_m}$ $\hfill$ attention scores
$\quad \boldsymbol{\alpha} \leftarrow \mathrm{Softmax}(\mathbf{s}, \mathrm{dim} = -1)$
$\quad \mathbf{r}_{\mathrm{mem}} \leftarrow \boldsymbol{\alpha}\mathcal{M}_{\mathrm{LT}}$
$\quad \mathbf{g} \leftarrow \sigma(\mathrm{Linear}_g([\mathbf{q}\|\mathbf{r}_{\mathrm{mem}}]))$ $\hfill$ gating
$\quad \mathbf{r}_{\mathrm{LT}} \leftarrow \mathbf{g} \odot \mathbf{r}_{\mathrm{mem}} + (1 - \mathbf{g}) \odot \mathbf{q}$
$\quad$ **return** $\mathbf{r}_{\mathrm{LT}}$

---

**Algorithm 10** Long-Term Branch Classifier

---

**Input** : $\mathbf{H}_{\mathrm{LT}} \in \mathbb{R}^{B \times N \times d_h}$; $\mathbf{p}_{\mathrm{LT}} \in \mathbb{R}^{B \times d_s}$: graph statistics; $\mathbf{r}_{\mathrm{LT}} \in \mathbb{R}^{B \times d_m}$: memory features
**Output** : $s^{\mathrm{LT}} \in \mathbb{R}^{B \times 2}$: class logits
$\mathbf{g} \leftarrow \mathrm{mean}(\mathbf{H}_{\mathrm{LT}}, \mathrm{dim} = 1)$ $\hfill$ graph embedding
$\quad \mathbf{h}_{\mathrm{LT}} \leftarrow [\mathbf{g}\|\mathbf{r}_{\mathrm{LT}}\|\mathbf{p}_{\mathrm{LT}}]$
$\quad \mathbf{x} \leftarrow \mathrm{Dropout}(\mathrm{ReLU}(\mathrm{Linear}_1(\mathbf{h}_{\mathrm{LT}})))$
$\quad \mathbf{x} \leftarrow \mathrm{Dropout}(\mathrm{ReLU}(\mathrm{Linear}_2(\mathbf{x})))$
$\quad s^{\mathrm{LT}} \leftarrow \mathrm{Linear}_{\mathrm{out}}(\mathbf{x})$
$\quad$ **return** $s^{\mathrm{LT}}$

---

## D.4. Feature Fusion

---

**Algorithm 11** Feature Fusion

---

**Input** : $\mathbf{h}_{\mathrm{LT}} \in \mathbb{R}^{B \times d_{\mathrm{LT}}}$: long-term features; $\mathbf{h}_{\mathrm{ST}} \in \mathbb{R}^{B \times d_{\mathrm{ST}}}$: short-term features; $s^{\mathrm{LT}}, s^{\mathrm{ST}} \in \mathbb{R}^{B \times 2}$: branch logits
**Output** : $s_{\mathrm{fus}} \in \mathbb{R}^{B \times 2}$: fused predictions
**if** *fusion_type* = *"simple_avg"* **then**
$\quad | \quad s_{\mathrm{fus}} \leftarrow (s^{\mathrm{LT}} + s^{\mathrm{ST}})/2$

**else if** *fusion_type* = *"weighted_linear"* **then**
$\quad | \quad \mathbf{w} \leftarrow \mathrm{Softmax}([\mathrm{Linear}_{\mathrm{LT}}(\mathbf{h}_{\mathrm{LT}}), \mathrm{Linear}_{\mathrm{ST}}(\mathbf{h}_{\mathrm{ST}})])$
$\quad | \quad s_{\mathrm{fus}} \leftarrow \mathbf{w}_{:,0} \cdot s^{\mathrm{LT}} + \mathbf{w}_{:,1} \cdot s^{\mathrm{ST}}$

**else if** *fusion_type* = *"nonlinear_ffn"* **then**
$\quad | \quad \mathbf{h}_{\mathrm{fus}} \leftarrow [\mathbf{h}_{\mathrm{LT}}\|\mathbf{h}_{\mathrm{ST}}]$
$\quad | \quad \mathbf{x} \leftarrow \mathrm{Dropout}(\mathrm{ReLU}(\mathrm{Linear}_1(\mathbf{h}_{\mathrm{fus}})))$
$\quad | \quad \mathbf{x} \leftarrow \mathrm{Dropout}(\mathrm{ReLU}(\mathrm{Linear}_2(\mathbf{x})))$
$\quad | \quad s_{\mathrm{fus}} \leftarrow \mathrm{Linear}_{\mathrm{out}}(\mathbf{x})$

**return** $s_{\mathrm{fus}}$

---

**Algorithm 12** Complete Model – Long-Term Processing

---

**Input** : $\mathbf{A}_{\mathrm{LT}}$: static connectivity; $\mathbf{X}_{\mathrm{coords}}$: coordinates; $\mathbf{C}_{\mathrm{LT}}$: communities
**Output** : $\mathbf{h}_{\mathrm{LT}} \in \mathbb{R}^{B \times d_{\mathrm{LT}}}$: long-term features; $s^{\mathrm{LT}} \in \mathbb{R}^{B \times 2}$: long-term logits
**// Long-Term Branch Processing**
$\quad \mathbf{H}_{\mathrm{LT}} \leftarrow \mathrm{LongTermGCNEncoder}(\mathbf{A}_{\mathrm{LT}}, \mathbf{X}_{\mathrm{coords}}, \mathbf{C}_{\mathrm{LT}})$
$\quad \mathbf{p}_{\mathrm{LT}} \leftarrow \mathrm{LongTermGraphStats}(\mathbf{A}_{\mathrm{LT}}, \mathbf{H}_{\mathrm{LT}})$
$\quad \mathbf{r}_{\mathrm{LT}} \leftarrow \mathrm{LongTermMemory}(\mathbf{H}_{\mathrm{LT}}, \mathcal{M}_{\mathrm{LT}})$
$\quad s^{\mathrm{LT}} \leftarrow \mathrm{LongTermClassifier}(\mathbf{H}_{\mathrm{LT}}, \mathbf{p}_{\mathrm{LT}}, \mathbf{r}_{\mathrm{LT}})$
$\quad \mathbf{h}_{\mathrm{LT}} \leftarrow [\mathrm{MeanPool}(\mathbf{H}_{\mathrm{LT}})\|\mathbf{p}_{\mathrm{LT}}\|\mathbf{r}_{\mathrm{LT}}]$
$\quad$ **return** $\mathbf{h}_{\mathrm{LT}}, s^{\mathrm{LT}}$

---

---

**Algorithm 13** Complete Model – Short-Term Processing

---

**Input** : $\mathbf{A}_{ST}$: temporal sequence; $\mathbf{X}_{coords}$: coordinates; $\mathbf{C}_{ST}$: communities; l: lengths; $\boldsymbol{\pi}$: community frequency
**Output** : $\mathbf{h}_{ST} \in \mathbb{R}^{B \times d_{ST}}$: short-term features; $s^{ST} \in \mathbb{R}^{B \times 2}$: short-term logits
**// Short-Term Branch Processing**
 $\{\mathbf{z}^{(m)}\}_{m=1}^M, \mathbf{p}_{ST} \leftarrow \text{TemporalDependencyEncoder-I}(\mathbf{A}_{ST}, \mathbf{X}_{coords}, \mathbf{C}_{ST}, l, \boldsymbol{\pi})$
 $\mathbf{y}_{cls} \leftarrow \text{TemporalDependencyEncoder-II}(\{\mathbf{z}^{(m)}\}_{m=1}^M, l)$
 $\mathbf{r}_{ST} \leftarrow \text{BrainFunctionMemory}(\mathbf{y}_{cls}, \mathcal{M}_{ST})$
 $s^{ST} \leftarrow \text{ShortTermClassifier}(\mathbf{y}_{cls}, \mathbf{r}_{ST}, \mathbf{p}_{ST})$
 $\mathbf{h}_{ST} \leftarrow [\mathbf{y}_{cls} \| \mathbf{r}_{ST} \| \mathbf{p}_{ST}]$
 **return** $\mathbf{h}_{ST}, s^{ST}$

---

**Algorithm 14** Complete Model – Final Fusion

---

**Input** : $\mathbf{h}_{LT}, s^{LT}$: long-term features and logits; $\mathbf{h}_{ST}, s^{ST}$: short-term features and logits
**Output** : $s_{pred} \in \mathbb{R}^{B \times 2}$: final predictions
**// Dual-Branch Fusion**
 $s_{pred} \leftarrow \text{DualBranchFusion}(\mathbf{h}_{LT}, \mathbf{h}_{ST}, s^{LT}, s^{ST})$
 **return** $s_{pred}$

---

### D.5. Notation Summary

*Table 3.* Summary of notation used in the algorithms.

| Symbol | Description | Symbol | Description |
|---|---|---|---|
| $B$ | Batch size | $N$ | Number of brain ROIs |
| $M$ | Number of time windows | $d, d_h$ | Hidden dimensions |
| $d_m$ | Memory dimension | $d_s$ | Statistics dimension |
| $L$ | Number of GCN layers | $L_{trans}$ | Transformer layers |
| $K$ | Memory bank size | $w$ | Window size |
| $\mathbf{A}$ | Adjacency matrix | $\mathbf{H}$ | Hidden representations |
| $\mathbf{C}, \mathbf{c}$ | Community labels | $\mathcal{M}$ | Memory bank |
| $\mathbf{z}^{(m)}$ | Window-level embedding | $\mathbf{y}_{cls}$ | CLS token output |
| $\mathbf{r}$ | Memory readout | $\mathbf{p}$ | Graph statistics |
| $s$ | Classification scores | $\boldsymbol{\pi}$ | Community frequency |
| $\sigma(\cdot)$ | Sigmoid function | $\odot$ | Element-wise product |
| $\|$ | Concatenation | $\mathbf{I}_N$ | Identity matrix |

