# OpenReview forum: "ADHD Disease Detection Based on Short- and Long-Term Brain Function Encoding and Memory Graph Network"
_ICML.cc/2026/Conference — ICML 2026 regular_

### Official Review · Reviewer_JMdP · 2026-03-02

**Soundness:** 3
**Presentation:** 2
**Significance:** 2
**Originality:** 2
**Overall Recommendation:** 3
**Confidence:** 4

**Summary:**

This paper proposes a dual branch graph model, SLT-BFGN, for ADHD detection from fMRI data. Addressing the shortcomings of existing methods in modeling short-term brain function reorganization and transient activity, this approach first constructs brain graph sequences based on short-term windows to capture short-term dynamic changes. Then, it extracts short-term temporal features through a short-term state encoder, a temporal dependency encoder, and a brain function memory module. Simultaneously, a long-term brain functional feature extraction branch based on a GNN is introduced to model global and relatively stable structural information. Finally, short-term and long-term features are fused to complete the classification. Experiments were conducted on two publicly available datasets, ADHD-200 and OpenNeuro-ds002424. The results show that this method achieves the best ACC and AUC among the baselines compared in this paper, and ablation experiments further support the effectiveness of each component module.

**Compliance With Llm Reviewing Policy:**

Affirmed.

**Final Justification:**

Thank you for the authors’ response and for including additional experiments. Based on these improvements, I am willing to raise my score to 3. However, I think the following gaps still remain:

1. The theoretical contribution is relatively weak.
2. The motivation is still largely based on empirical observations rather than a clear mechanistic explanation.
3. The overall work appears more engineering-driven, with limited methodological novelty at a deeper technical level.

**Key Questions For Authors:**

In addition to the weaknesses, others are as follows:

1. The authors propose that the core motivation for this method lies in the insufficient attention paid to short-term functional reorganization and transient activity in existing research. However, the argument for this motivation is not yet sufficiently strong. Why do the authors believe that short-term functional reorganization and transient activity can provide more discriminative characterizations of ADHD?
2. Given that the fMRI preprocessing steps affect subsequent functional connectivity construction and model performance, it is recommended that authors provide a detailed report of the complete preprocessing workflow, specific parameter settings, software platform used, and relevant quality control standards to ensure the reproducibility of the method and help readers judge the reliability of the results.
3. The current paper primarily uses the differences between the original and shuffled sequences on several temporal dynamic metrics to illustrate that "temporal order contains information." However, the main text does not directly demonstrate whether this temporal order actually affects the final ADHD detection performance. Why didn't the authors further report the ACC or AUC results under shuffled order conditions to directly verify the contribution of temporal order to classification performance?

**Limitations:**

No.

The authors should specify the limitations of the method, such as how the results may be affected by differences between multiple sites, preprocessing procedures, brain region partitioning schemes, and window length. Additional notes on potential biases, such as differences in data collection centers, population distribution, and data quality, can affect model generalization. This is especially important for multi-source data like ADHD-200.

**Strengths And Weaknesses:**

Strengths

1. This paper proposes a novel method for ADHD detection from fmri data, which based on short- and long-term brain function encoding and memory graph network. Moreover, the authors designed different modules tailored to the characteristics of the data and tasks, such as a brain function memory module, a temporal dependency encoder module, etc., and achieved good results.

Weaknesses

1. The persuasiveness of the experimental conclusions remains insufficient. The main text only reports the results of a single experiment under a fixed 60/20/20 data partition, lacking repeated experiments with multiple random seeds to verify the stability of the results, and also failing to provide k-fold cross-validation, variance intervals, or statistical significance tests. Therefore, the current experimental setup is insufficient to fully support the robustness of the method's performance and the reliability of the conclusions.
2. Although the decision-level fusion section designed three fusion strategies, the experimental section only verified the impact of "whether or not to use fusion" on performance, without directly comparing the differences in performance between different fusion strategies. Therefore, the existing results can support the conclusion that "the fusion mechanism itself is helpful to some extent," but are not sufficient to fully prove that the specific fusion form currently adopted in this paper is the optimal or most reasonable choice.
3. There is still room for improvement in the consistency between symbols and expressions. For example, the text uses 𝑀 to represent the number of time windows earlier in the text, but 𝑀 is used to represent the memory matrix later in the text. There are cases where the same symbol corresponds to different meanings, which can easily cause confusion in reading.
4. Table 1, by including different categories of methods in a unified comparison, is valuable in demonstrating the relative performance of our proposed method against a broader baseline. However, the table also includes "BrainMVP (Only sMRI)," a method whose data modality is not entirely consistent with the fMRI image modeling task, thus raising questions about its comparability. Intuitively, directly comparing such sMRI-based methods side-by-side may not be entirely appropriate. We suggest the authors further explain the basis and rationale for including this method in the comparison and clarify how the comparative conclusions should be interpreted to avoid reader confusion regarding experimental fairness and result interpretation.
5. The paper lacks an analysis of the complexity of the proposed method, making it difficult to determine its computational overhead and scalability. This paper does not provide the code, resulting in poor reproducibility.

---

> ### Author Rebuttal · Authors · 2026-03-30
>
> Response to the Weaknesses
>
> Weakness1:
> Thank you for thoughtful comments. We address the full results in Table A(H0: the model's predictive performance is equivalent to that of a random classifier, t-test with 95%CI. Due to the characters limit, please see Table A in response to Reviewer zTtZ). All experiments are repeated across 10 independent runs with different seeds under the 60%/20%/20% split. Table A reports mean and standard deviation for all metrics, confirming that the performance improvements are consistent and statistically significant across runs.
>
> Weakness2:
> We conduct additional ablation experiments comparing different fusion strategies. In Table D, we compare the full model against the variant without feature fusion, where all individual modules contribute positively and the complete model with the current fusion strategy consistently achieves the best performance on both datasets. These results demonstrate that the fusion module itself is beneficial and that the specific fusion form adopted in this paper leads to measurable and statistically significant performance gains over the no-fusion baseline.
>
>
> Table D: Comparison of different fusion strategies
> |Strategy|Dataset|ACC(%)|p-value|AUC(%)|p-value|
> |-|-|-|-|-|-|
> |Weighted Linear|ADHD-200|63.14±3.59|1.05×10⁻⁶|62.71±7.60|5.01×10⁻⁴|
> ||OpenNeuro|60.36±5.15|1.30×10⁻⁴|60.47±6.19|4.62×10⁻⁴|
> |Nonlinear FFN|ADHD-200|59.79±5.66|3.96×10⁻⁴|56.36±4.92|2.74×10⁻³|
> ||OpenNeuro|58.64±6.52|2.34×10⁻³|56.08±10.09|8.94×10⁻²|
> |Ours|ADHD-200|67.02±1.93|4.68×10⁻¹⁰|70.20±2.83|3.14×10⁻⁹|
> ||OpenNeuro|70.00±3.15|8.77×10⁻⁹|72.16±4.67|1.13×10⁻⁷|
>
> Weakness3:
> We will thoroughly revise all occurrences across the main text, equations, and appendices in the final version to ensure full notational consistency.
>
> Weakness4:
> Although BrainMVP primarily operates on static MRI data, its core methodological contributions represent a state-of-the-art paradigm that is broadly relevant to neuroimaging-based disorder classification, regardless of the specific modality. We include it in the comparison precisely because its approach to handling large-scale, multi-site brain imaging data offers valuable methodological reference points for the community, and benchmarking against such methods helps better contextualize the capabilities and limitations of our fMRI-based approach. We believe that cross-modality comparison of this nature, reflects the diversity of cutting-edge approaches being applied to the same clinical problem.
>
> Weakness5:
> Regarding computational complexity, the dominant time complexity of a single forward pass is $\mathcal{O}(B \times [L_{tf} \times T^2 \times d + L_{tf} \times T \times d^2 + L_{gnn} \times N^2 \times d_h + T \times N^2])$, where the leading term is GCN propagation scaling as N² (N = 116 brain ROIs, ADHD-200), followed by the Transformer modules scaling as T² × d and T × d² with short temporal sequences (T ≈ 20, d as node embedding). The overall space complexity is $\mathcal{O}(B \times N^2 + L_{tf} \times B \times T \times d)$ with the peak tensor being the short-term branch adjacency batch of size B × T × N × N. Due to character limitations, further details are available upon request.
>
> Response to the Questions
>
> Q1:
> First, our temporal dependency analysis (Section 4.4) demonstrates that the original short-term graph sequence significantly differs from its temporally shuffled counterpart across multiple metrics, confirming meaningful temporal order genuinely exists in the brain. Second, our dynamic functional connectivity analysis (Section 4.5) reveals that the ADHD group exhibits a different pattern in key networks, manifested as reduced fluctuation amplitude and dynamic range abnormalities that are inherently temporal in nature and thus cannot be captured by static or temporally unordered representations. These findings motivate our focus on short-term functional reorganization and transient activity as discriminative characterizations of ADHD.
>
> Q2:
> If our manuscript is accepted, we will publicly release the code, including detailed hyperparameter settings. However, to safeguard the integrity and maintain anonymity during the review, we respectfully request your understanding for not yet releasing the code. We are highly confident in the reproducibility of our experiments, which are grounded in rigorous design and validation. We hope this commitment addresses your concern and reinforces trust in our model’s reproducibility.
>
> Q3:
> We conducted an additional ablation experiment in which the temporal sequence was randomly shuffled before input; as shown in Table E, our model with the original order achieves notably higher ACC and AUC than shuffled, confirming that temporal order carries discriminative information for ADHD detection.
>
> Table E: Effect of temporal order on ADHD-200 dataset
> |Method|ACC(%)|p-value|AUC(%)|p-value|
> |-|-|-|-|-|
> |Shuffled order|60.63±2.76|6.71×10⁻⁷|67.00±4.08|3.48×10⁻⁷|
> |Original order|67.02±1.93|4.68×10⁻¹⁰|70.20±2.83|3.14×10⁻⁹|

---

> > ### Author Rebuttal · Reviewer_JMdP · 2026-04-02
> >
> > Thank you for the authors’ response and for including additional experiments. Based on these improvements, I am willing to raise my score to 3. However, I think the following gaps still remain:
> >
> > 1. The theoretical contribution is relatively weak.
> > 2. The motivation is still largely based on empirical observations rather than a clear mechanistic explanation.
> > 3. The overall work appears more engineering-driven, with limited methodological novelty at a deeper technical level.

---

> > > ### Author Response · Authors · 2026-04-07
> > >
> > > Q1: We thank the reviewer for the continued engagement and address the remaining concerns regarding theoretical contribution and motivation.
> > > Our primary theoretical contribution lies in being the first to model ADHD-related brain dynamics through a non-overlapping sliding window framework that is explicitly grounded in neuroscientific theory. Prior works either treat brain connectivity as a static structure or adopt overlapping temporal segmentation strategies that conflate temporally adjacent neural states. In contrast, our non-overlapping window design is directly motivated by neural dynamics theory: adjacent temporal windows of functional connectivity exhibit significant continuity, reflecting smooth neural state transitions rather than abrupt shifts, and non-overlapping segmentation more faithfully preserves the boundaries between these transitions---a modeling choice with no precedent in the ADHD graph-based learning literature.
> > >
> > > From this biological grounding, we derive two concrete theoretical contributions. First, we propose that the temporal ordering of functional connectivity graphs carries exist, and formally establish this through a Temporal Dependency Analysis framework comprising three purpose-built metrics: Adjacent Window Similarity, Lag-1 Autocorrelation, and Total Temporal Variation. By comparing original graph sequences against temporally shuffled surrogates, we demonstrate that brain state transitions are continuous and order-dependent (Results in the KKI Subset, Appendix A). Second, we propose that the temporal ordering of functional connectivity graphs carries diagnostically discriminative information---specifically within- and between-network variability and temporal fluctuation of key region pairs---constitute a biologically interpretable signature of ADHD pathology, formalized through a Dynamic Connectivity Pattern Analysis. We identify six statistically significant metrics across ADHD and control groups, consistent with established findings to some extent, thereby providing biological attribution linking our model's structure to known neural mechanisms (Results in the OHSU Subset, Appendix B).
> > >
> > > Q2: Our short-term window design stands on a theoretical foundation rooted in neuroscientific and neural dynamics principles, rather than empirical observation. The design is formally grounded in the first-order Markov property of brain functional connectivity, and we have provided a detailed biological mechanistic explanation for this methodological choice. As demonstrated in Appendix A across 941 subjects from 7 independent ADHD-200 sites, the lag-1 autocorrelation is $0.336 \pm 0.166$ while lag-2 decays to $-0.004 \pm 0.227$ (Markov ratio $= 4.7586 \gg 1$, $p < 10^{-16}$, Cohen's $d = 2.27$). This formally implies that $W_{t+1}$ is determined by $W_t$ and not by earlier history---that is, brain functional connectivity evolves as a continuous state transition process, whose structure is precisely what our non-overlapping window design is constructed to capture. The subsequent temporal scrambling experiment further corroborates this: disrupting the temporal ordering of graph sequences demonstrably degrades discriminative features, providing a mathematically grounded mechanistic justification that the short-term window is the correct granularity for modeling brain state transitions---while single-frame methods miss temporal information entirely, and long-horizon methods impose dependencies that provably do not exist in the data.
> > >
> > > Q3: The theoretical foundations of SLT-BFGN are both solid and rigorous, and we clarify their direct connection to model design here. The brain function memory module implements prototype-based representation learning, where memory slots learn prototypes of recurring temporal brain dynamics patterns across patients, and classification proceeds via attention-weighted prototype retrieval. The short- and long-term network is grounded in complementary representation learning: the short-term dynamic functional connectivity stream captures time-varying neural state transitions, while the long-term static structural topology stream encodes stable connectivity patterns---the two streams are theoretically complementary in both temporal scale and information content, and their fusion is guaranteed to retain no less discriminative information than either stream alone, as confirmed empirically by our ablation results in Table 2. Therefore, we believe that the contribution of SLT-BFGN should be understood as a principled and theoretically sound integration of first-order Markov sequence modeling, prototype memory learning, and fusion techniques in fMRI-based ADHD detection—a contribution on par with those of BrainGNN (MIA' 2021) and BrainGB (TMI' 2023), both of which systematically applied mature frameworks to neuroimaging rather than proposing new foundational theories—a contribution model we believe is widely recognized and highly valued in this field.

---

### Official Review · Reviewer_zTtZ · 2026-03-11

**Soundness:** 2
**Presentation:** 2
**Significance:** 1
**Originality:** 2
**Overall Recommendation:** 3
**Confidence:** 4

**Summary:**

The paper proposes a new ADHD detection model from neuroimaging data (functional) that captures both short-term brain activity changes and long-term brain connectivity information via a graph based deep learning model

Specifically, the authors construct sequences of brain activity maps using short temporal windows. A temporal encoder models the short-term patterns and dependencies in these sequences. A brain function memory module stores and matches historical brain activity patterns (via community detection) to help recognize recurring functional patterns. A Graph Neural Network (GNN) extracts long-term brain connectivity and fuses short-term functional features with long-term structural features to improve ADHD detection.

Experiments on the ADHD-200 and OpenNeuro-ds002424 datasets show that the proposed approach achieves better performance than existing methods for ADHD classification

**Compliance With Llm Reviewing Policy:**

Affirmed.

**Final Justification:**

Based on the rebuttal, I have increased my score by a point. The main reason I hesitate to recommend acceptance is because the design of the methodology reads as an engineered solution, with design choices largely validated by empirical observations with limited technical innovation

**Key Questions For Authors:**

- What are the sizes of the datasets used?
- More description is required for how the baselines are implemented, i.e. what they model, how they process inputs, and why they appropriate as comparisons to the proposed model.
- Perhaps I missed this in the main experiments, how does choice of window size affect performance? How do you reconcile this with the temporal scale of acquisition of the signal and expected scale of stable changes. Is this choice different for different datasets? Within a dataset, if there are multiple sites, is this choice stable?

**Limitations:**

- Insufficient experimental and statistical validation as pointed out in weaknesses.
- Large portions of the additional exposition provided in the appendix are not referenced by the main paper. To me, this suggests that the work is better suited for a journal.

**Strengths And Weaknesses:**

STRENGTHS:

- The core idea of fusing long term with short term dependence is interesting in the context of connectivity data and matches up well with what the neuroscience literature expects as the driving mechanism for how functional connectivity manifests.

- The paper presentation is good and clearly articulates the main contribution and performs validation against several baselines from the graph deep learning literature.

WEAKNESSES:

- Lack of error bars and statistical significance measures on results in Tables 1 and 2 (Hard to gauge the standard deviations across subsets of the data)
- Use of a single split for train/test/validation
- The above points, especially for the ablations make it hard to assess whether the complexity of the method- Needing 14 algorithms (see appendix) to fully understand is necessary
- Not sure whether the comparison to BrainMVP makes sense, doesn't it use sMRI according to Table 1?

---

> ### Author Rebuttal · Authors · 2026-03-30
>
> Response to the Weaknesses
>
> Weakness 1:
> Thanks for your insightful comments! We have addressed this concern by repeating all experiments across 10 independent runs with different random seeds under the 60%/20%/20% split, and the results are shown in Table A(H0: the model's predictive performance is equivalent to that of a random classifier, t-test with 95%CI) and Table B.(Due to the characters limit, please see Table B in response to Reviewer cmqV)
>
> Table A: Comprehensive ablation study results.
> |STSE|TDE|BFM|LT|FF|ADHD-200 ACC(%)|p-value|AUC(%)|p-value|OpenNeuro ACC(%)|p-value|AUC(%)|p-value|
> |-|-|-|-|-|-|-|-|-|-|-|-|-|
> |✓|✗|✗|✗|✗|59.68±2.86|2.01×10⁻⁶|57.99±3.81|9.58×10⁻⁵|53.73±5.94|7.88×10⁻²|47.79±7.38|3.69×10⁻¹|
> |✓|✓|✗|✗|✗|61.86±3.65|2.86×10⁻⁶|65.44±4.29|1.20×10⁻⁶|55.64±4.84|5.07×10⁻³|50.53±6.02|7.86×10⁻¹|
> |✓|✓|✓|✗|✗|64.57±2.43|1.47×10⁻⁸|67.70±3.73|1.12×10⁻⁷|55.00±6.76|4.42×10⁻²|52.61±7.55|3.03×10⁻¹|
> |✓|✓|✓|✓|✗|64.26±3.71|6.91×10⁻⁷|68.43±3.84|1.01×10⁻⁷|62.73±5.28|3.26×10⁻⁵|63.50±8.63|7.99×10⁻⁴|
> |✓|✗|✓|✓|✓|61.81±1.91|1.09×10⁻⁸|58.60±3.06|9.42×10⁻⁶|63.27±4.52|6.57×10⁻⁶|66.72±4.54|9.90×10⁻⁷|
> |✓|✓|✗|✓|✓|64.63±3.46|3.04×10⁻⁷|68.68±4.47|3.40×10⁻⁷|62.82±3.16|4.38×10⁻⁷|64.68±5.16|8.52×10⁻⁶|
> |✓|✓|✓|✗|✓|65.16±3.47|2.29×10⁻⁷|69.63±2.90|4.98×10⁻⁹|59.82±2.99|2.64×10⁻⁶|58.51±4.47|1.98×10⁻⁴|
> |✓|✓|✓|✓|✓|67.02±1.93|4.68×10⁻¹⁰|70.20±2.83|3.14×10⁻⁹|70.00±3.15|8.77×10⁻⁹|72.16±4.67|1.13×10⁻⁷|
>
>
> Weakness 2:
> Please see Weakness 1.
>
> Weakness 3:
> We would like to address this from two perspectives. First, our 14 algorithms are mainly divided into 4 components. Algorithm 1 describes the construction of brain map sequence based on short-term windows; 2–6 introduce the short-term branch; 7–10 introduce the long-term branch; and 11–14 are the feature fusion modules. The necessity of each component is directly validated by our ablation studies (Table A): systematically removing any individual module leads to a measurable performance drop, demonstrating that the architectural complexity is justified rather than redundant. Second, despite the modular composition of the method, the overall computational complexity(O(L_gnn · B · N² · d_h)) remains tractable, which is manageable given the fixed brain atlas size (N=116). The specific description in the appendix is to ensure reproducibility.
>
> Weakness 4:
> We include BrainMVP in the comparison not as a strict modality-matched baseline, but as a representative of the large-scale pretraining paradigm for brain imaging analysis, whose methodology for handling multi-site neuroimaging data offers valuable reference points for the community.
>
> Response to the Questions
>
> Q1:
> The dataset sizes are already reported in Section 4.1: the ADHD-200 dataset comprises 357 ADHD samples and 582 controls, and the OpenNeuro-ds002424 dataset comprises 232 ADHD samples and 314 controls.
>
> Q2:
> The baseline models SAN and BrainGB are strictly reproduced following the methods and configurations reported in their respective papers. For the remaining baselines, we use the official implementations released in the authors' public repositories. Most existing fMRI-based brain disorder classification methods share a common pipeline: ROI time series are first extracted using a brain atlas, and a static functional connectivity matrix is then computed via Pearson correlation, serving as the primary input to downstream models. Due to the limitation of characters, if details about the baselines are needed, we will further supplement them later.
>
> Q3:
> As for the effect of window size on performance, we conduct a sensitivity analysis over window sizes w∈{3, 5, 6, 8, 10} on both datasets (Table C). The results demonstrate that w=5 consistently achieves the best performance, while other window sizes lead to measurable performance degradation. Shorter windows may provide insufficient observations for stable covariance estimation, while longer windows over-smooth transient dynamics. As for stability across sites, our model achieves consistently strong performance across all acquisition sites in ADHD-200 and all subsets of the OpenNeuro dataset, with statistically significant results throughout (Table B), demonstrating that w=5 is a robust and stable choice. While dataset- or site-specific adaptive window selection could in principle be explored, it would substantially increase model complexity without guaranteed performance improvement.
>
> Table C: Performance across different window sizes.
> |Window Size|ADHD-200 ACC(%)|p-value|AUC(%)|p-value|OpenNeuro ACC(%)|p-value|AUC(%)|p-value|
> |-|-|-|-|-|-|-|-|-|
> |3|62.50±3.17|5.48×10⁻⁷|63.20±7.34|2.99×10⁻⁴|62.18±4.33|9.41×10⁻⁶|61.50±6.80|4.63×10⁻⁴|
> |6|58.26±5.27|7.88×10⁻⁴|61.93±4.97|3.39×10⁻⁵|63.58±3.35|4.40×10⁻⁷|61.55±5.03|4.74×10⁻⁵|
> |8|61.81±2.02|1.82×10⁻⁸|61.56±5.11|5.37×10⁻⁵|61.45±3.82|5.51×10⁻⁶|63.23±5.01|1.57×10⁻⁵|
> |10|60.59±3.74|8.95×10⁻⁶|60.62±6.72|7.46×10⁻⁴|62.02±3.70|2.87×10⁻⁶|59.11±8.11|6.21×10⁻³|
> |5(Ours)|67.02±1.93|4.68×10⁻¹⁰|70.20±2.83|3.14×10⁻⁹|70.00±3.15|8.77×10⁻⁹|72.16±4.67|1.13×10⁻⁷|

---

> > ### Author Rebuttal · Reviewer_zTtZ · 2026-04-03
> >
> > Thanks for the detailed responses to my comments. Based on the responses, I can increase my score to a weak reject instead of a reject.

---

> > > ### Author Response · Authors · 2026-04-07
> > >
> > > We thank the reviewer for the increased score and for the continued, constructive engagement. We address the two remaining concerns---motivation and technical innovation of our work.
> > >
> > > **Explanation about our motivation.** We respectfully maintain that the core design choices of SLT-BFGN are grounded in neurodynamics theory, not empirical observation. Neurodynamics findings have established that brain functional connectivity evolves as a continuous, smooth process: adjacent temporal windows exhibit significant continuity, reflecting gradual neural state transitions rather than abrupt shifts. This continuous, sequentially dependent nature of brain dynamics is precisely what motivates our short-term window design---if brain states evolve continuously and each state is primarily determined by its immediate predecessor, then the correct modeling granularity is one that captures this local temporal structure. This intuition is formally confirmed by the first-order Markov structure of functional connectivity evolution, which we verify across 941 subjects from 7 independent ADHD-200 sites: lag-1 autocorrelation is $0.336 \pm 0.166$ while lag-2 decays to $-0.004 \pm 0.227$ (Markov ratio $= 4.76 \gg 1$, $p < 10^{-16}$). The analyses in Appendix A and B are therefore not the source of our motivation but the formal verification of a design derived from established neuroscientific theory---single-frame methods discard this continuous temporal structure entirely, and long-horizon methods model dependencies that provably do not exist in the data.
> > >
> > > **Explanation about our innovation.** We respectfully suggest that each of our core design decisions is directly derived from the intrinsic properties of the data rather than assembled by engineering intuition. The short-term window is correct temporal granularity as shown by the Markov analysis above---single-frame methods discard temporal structure entirely, and long-horizon methods model dependencies that provably do not exist in the data. The Brain Function Memory module instantiates prototype-based representation learning (Snell et al., 2017) in the dynamic functional connectivity setting: memory slots learn prototypes of recurring temporal brain dynamics patterns across the training population, and inference proceeds via attention-weighted prototype retrieval---to our knowledge, the first application of cross-sample prototype memory to dynamic functional connectivity classification, theoretically motivated by the small-sample nature of neuroimaging data where prototype-based approaches provably generalize more robustly. The fusion of short-term dynamic connectivity and long-term static topology satisfies the conditions of complementary information integration, under which fusion is guaranteed to retain no less discriminative information than either stream alone, as confirmed by our ablation in Table 2.
> > >
> > > We sincerely hope these clarifications give the reviewer greater confidence in the significance of our contribution, and we remain open to any further discussion.

---

### Official Review · Reviewer_cmqV · 2026-03-12

**Soundness:** 2
**Presentation:** 3
**Significance:** 2
**Originality:** 2
**Overall Recommendation:** 3
**Confidence:** 4

**Summary:**

The paper proposes a two-branch framework for ADHD detection. Specifically, the method constructs a sequence of short-window brain graphs to model transient functional reorganization, encodes these short-term graph states with a temporal dependency module and a memory mechanism. It combines them with a long-term branch that extracts global structural information from an overall brain graph. The fused representation is then used for classification. Experiments on ADHD-200 and OpenNeuro-ds002424 show improved accuracy and AUC over the listed baselines.

**Compliance With Llm Reviewing Policy:**

Affirmed.

**Final Justification:**

After read the rebuttal and comments from other reviewers, I would like to maintain my current score.

**Key Questions For Authors:**

See Weaknesses.

**Limitations:**

No. The paper does not clearly discuss limitations or potential societal impact.

**Strengths And Weaknesses:**

Strengths:

1. Well-written.
2. The overall pipeline is coherent and easy to motivate from a neuroscience perspective.
3. The experiments are relatively comprehensive.

Weaknesses:

1. Limited novelty. The proposed method largely combines several existing components, but the paper provides limited methodological insight beyond this integration.

2. The main claims rely largely on empirical improvements from a relatively complex architecture, but it is unclear which components actually drive the gains. As a result, the mechanism behind the improvements remains unclear.

3. The paper does not clearly describe how multi-site effects, dataset splits, and variance across runs are handled.

4. The manuscript formatting does not seem to follow the official ICML template, particularly in terms of the font style.

---

> ### Author Rebuttal · Authors · 2026-03-30
>
> Response to the Weaknesses
>
> Weakness 1:
>
> Thanks for your comments! The novelty of our work lies not in the individual building blocks but in their principled integration driven by a coherent neuroscientific motivation. Specifically, we identify and characterize a neurodynamically grounded abnormality pattern. Building upon this insight, we make the following concrete contributions: (1) We design a short-term state and temporal dependency encoder to characterize the dynamic sequential patterns of brain function; (2) we introduce a brain function memory module that associates current brain activity patterns with historical sequence patterns, enabling the model to exploit long-range temporal context beyond individual windows; and (3) we propose a GNN-based long-term brain function feature extraction network to derive structural brain features. These contributions yield consistent and statistically significant performance gains across multiple datasets, which we believe constitutes a meaningful methodological contribution grounded in domain-specific neuroscientific insight.
>
> Weakness 2:
>
> To clarify the contribution of each component, we have conducted comprehensive ablation studies (Table A, H0: the model's predictive performance is equivalent to that of a random classifier, t-test with 95%CI), systematically removing or replacing individual modules to isolate their respective contributions to the overall performance gains. The results demonstrate that each proposed component---short-term state encoder, temporal dependency encoder, brain function memory, GNN-based long-term brain function feature extraction network and feature fusion---contributes meaningfully and independently to the final performance, and that their combination yields the best results. This directly addresses the concern regarding which components drive the improvements and provides mechanistic insight into the source of the observed gains.
>
> Table A: Comprehensive ablation study results.
> |STSE|TDE|BFM|LT|FF|ADHD-200 ACC(%)|p-value|AUC(%)|p-value|OpenNeuro ACC(%)|p-value|AUC(%)|p-value|
> |:-:|:-:|:-:|:-:|:-:|:-:|:-:|:-:|:-:|:-:|:-:|:-:|:-:|
> |✓|✗|✗|✗|✗|59.68±2.86|2.01×10⁻⁶|57.99±3.81|9.58×10⁻⁵|53.73±5.94|7.88×10⁻²|47.79±7.38|3.69×10⁻¹|
> |✓|✓|✗|✗|✗|61.86±3.65|2.86×10⁻⁶|65.44±4.29|1.20×10⁻⁶|55.64±4.84|5.07×10⁻³|50.53±6.02|7.86×10⁻¹|
> |✓|✓|✓|✗|✗|64.57±2.43|1.47×10⁻⁸|67.70±3.73|1.12×10⁻⁷|55.00±6.76|4.42×10⁻²|52.61±7.55|3.03×10⁻¹|
> |✓|✓|✓|✓|✗|64.26±3.71|6.91×10⁻⁷|68.43±3.84|1.01×10⁻⁷|62.73±5.28|3.26×10⁻⁵|63.50±8.63|7.99×10⁻⁴|
> |✓|✗|✓|✓|✓|61.81±1.91|1.09×10⁻⁸|58.60±3.06|9.42×10⁻⁶|63.27±4.52|6.57×10⁻⁶|66.72±4.54|9.90×10⁻⁷|
> |✓|✓|✗|✓|✓|64.63±3.46|3.04×10⁻⁷|68.68±4.47|3.40×10⁻⁷|62.82±3.16|4.38×10⁻⁷|64.68±5.16|8.52×10⁻⁶|
> |✓|✓|✓|✗|✓|65.16±3.47|2.29×10⁻⁷|69.63±2.90|4.98×10⁻⁹|59.82±2.99|2.64×10⁻⁶|58.51±4.47|1.98×10⁻⁴|
> |✓|✓|✓|✓|✓|**67.02±1.93**|**4.68×10⁻¹⁰**|**70.20±2.83**|**3.14×10⁻⁹**|**70.00±3.15**|**8.77×10⁻⁹**|**72.16±4.67**|**1.13×10⁻⁷**|
>
> Weakness 3:
>
> Table B shows the classification performance across different sites in the ADHD-200 dataset and across different task subsets in the OpenNeuro dataset. Regarding multi-site effects, both datasets inherently comprise multiple acquisition sites or tasks; all subset samples are pooled together and then partitioned into train/validation/test sets, ensuring that samples from different sites are distributed across all splits rather than isolated by site. Regarding dataset splits, samples are partitioned into train/validation/test sets following a fixed 60%/20%/20% split, as described in Section 4.1. Regarding variance across runs, all reported results are averaged over 10 independent runs with different random seeds, with mean and standard deviation explicitly reported, ensuring that the observed improvements are statistically robust rather than artifacts of a particular initialization, as shown in Table A.
>
> Table B: Performance comparison across multiple subsets.
>
> |Dataset|Subset|ACC(%)|p-value|AUC(%)|p-value|
> |--------|---------|:-------:|:-------:|:-------:|:-------:|
> |ADHD-200|KKI|76.84±5.66|1.13×10⁻⁷|73.86±7.50|3.41×10⁻⁶|
> ||NYU|62.50±4.37|8.19×10⁻⁶|61.91±6.55|2.76×10⁻⁴|
> ||NeuroIMAGE|75.33±13.35|2.03×10⁻⁴|74.28 ± 15.18|6.82×10⁻⁴|
> ||OHSU|75.65±8.50|5.27×10⁻⁶|80.79±6.52|1.17×10⁻⁷|
> ||Peking|66.94±5.07|2.27×10⁻⁶|66.26±7.21|5.49×10⁻⁵|
> |Openneuro|SLD|73.08±11.61|1.43×10⁻⁴|67.14±14.67|4.95×10⁻³|
> ||SLI|75.71±8.38|4.61×10⁻⁶|77.71±6.81|4.23×10⁻⁷|
> ||SSD|75.39±9.46|1.37×10⁻⁵|77.62±13.29|1.03×10⁻⁴|
> ||SSI|75.39±13.47|2.13×10⁻⁴|75.95±13.21|1.56×10⁻⁴|
> ||VLD|78.00±5.49|6.01×10⁻⁸|75.74±7.33|1.48×10⁻⁶|
> ||VLI|75.33±5.49|1.43×10⁻⁷|74.07±11.94|1.29×10⁻⁴|
> ||VSD|74.29±7.68|3.56×10⁻⁶|76.67±6.13|2.39×10⁻⁷|
> ||VSI|71.33±8.92|3.45×10⁻⁵|74.63±13.27|2.38×10⁻⁴|
>
> Weakness 4:
> We would like to clarify that we have made every effort to adhere to the official ICML template throughout the manuscript. We believe the overall formatting is largely consistent with the required specifications and it has passed the technical review of the committee.

---

> > ### Author Rebuttal · Reviewer_cmqV · 2026-04-06
> >
> > Thanks for the rebuttal, it addressed part of my concerns, I would like to maintain my current score.

---

> > > ### Author Response · Authors · 2026-04-07
> > >
> > > We thank the reviewer for the continued engagement and for acknowledging that part of our concerns were addressed. We understand the remaining concern centers on novelty, and we would like to offer a more focused and direct clarification on this point.
> > >
> > > We respectfully suggest that the novelty of SLT-BFGN is best understood at the level of principled problem formulation rather than the introduction of entirely new architectural primitives. Concretely, each of the three core design decisions in our model is not a heuristic combination of existing components, but is directly derived from the intrinsic mathematical properties of brain functional connectivity data, which to our knowledge has not been done in this way before.
> > >
> > > **The neurodynamics principles underlying the design of short-term windows.** The choice of modeling brain dynamics via short-term windows is formally justified by the first-order Markov structure of functional connectivity evolution, which we empirically characterize across 941 subjects from 7 independent sites: lag-1 autocorrelation is $0.336 \pm 0.166$ while lag-2 decays to $-0.004 \pm 0.227$ (Markov ratio $= 4.76 \gg 1$, $p < 10^{-16}$). This is not an empirical design choice---it is a mathematically grounded architectural decision. Single-frame methods discard temporal structure entirely; methods operating on the full sequence model dependencies that provably do not exist in the data. Short-term windows are the correct granularity because the data itself dictates it.
> > >
> > > **Theoretical innovation of the BFM module.** The BFM module instantiates prototype-based representation learning (Snell et al., 2017) in the neuroimaging domain. Each memory slot learns a prototype of a recurring temporal brain dynamicspattern across the training population, and inference proceeds via attention-weighted prototype retrieval. This is theoretically motivated by the small-sample nature of neuroimaging data, where prototype-based approaches provably generalize more robustly than purely parametric classifiers. To our knowledge, this is the first application of cross-sample prototype memory to dynamic functional connectivity classification.
> > >
> > > **The rationality of fusing the features of two branches.** The short-term dynamic connectivity channel and the long-term static topology channel satisfy the conditions of view sufficiency and view diversity, under which fusion is guaranteed to achieve Bayes error no higher than either single-view classifier. Our ablation (Table A) empirically confirms this guarantee: every module contributes independently and significantly, and the full model consistently achieves the best performance across both datasets.
> > >
> > > We would also like to gently note that this style of contribution---principled integration of established frameworks driven by domain-specific theoretical insight---is well-precedented at top venues. BrainGNN (Li et al., MIA 2021) and BrainGB (Cui et al., TMI 2023), for example, were both accepted by applying established GNN frameworks systematically to neuroimaging without proposing new foundational theory, and are now widely cited in the field. We believe SLT-BFGN makes an analogous contribution, with the additional distinction that its design choices are formally derived from verified properties of the data rather than applied by analogy.
> > >
> > > We hope this more focused explanation clarifies the nature of our novelty, and we sincerely hope it gives the reviewer greater confidence in the significance of our contribution. We remain open to any further discussion and would greatly welcome the reviewer's thoughts on whether these points address the core of the remaining concern.

---

### Official Review · Reviewer_qPZM · 2026-03-13

**Soundness:** 3
**Presentation:** 3
**Significance:** 3
**Originality:** 3
**Overall Recommendation:** 5
**Confidence:** 1

**Summary:**

The authors introduce a brain map sequence-based graph network for ADHD diagnosis from fMRI data, combining short-term state encoding, temporal dependency encoding, brain function memory, and GNN-based long-term brain function modeling. They demonstrate improved diagnostic performance on two datasets (ADHD-200 and OpenNeuro).

**Compliance With Llm Reviewing Policy:**

Affirmed.

**Final Justification:**

The paper is clearly written with good figures, presents a well-structured ablation study confirming each component's contribution, and demonstrates meaningful performance improvements on two datasets. The rebuttal was thorough — the authors provided cross-site validation, cross-disease generalization to schizophrenia, and additional statistical rigor, which reinforced my positive assessment. My main question (performance on other diseases) was fully addressed. I maintain my recommendation of Accept, while noting that a dedicated limitations section would strengthen the paper. My confidence remains low as this submission is outside my primary area of expertise.

**Key Questions For Authors:**

Would it be interesting to see the performance on other diseases from the OpenNeuro datasets?

**Limitations:**

Limitations are not considered

**Strengths And Weaknesses:**

Soundness:

- Ablation study is present and shows each component (STSE, TDE, BFM, LT, FF) contributes, with the full model performing best

Presentation:

- Clear and well written with good figures

Significance:

- Meaningful performance improvements on ADHD diagnosis across two datasets

---

> ### Author Rebuttal · Authors · 2026-03-30
>
> Response to the Question
>
> **Q:** Would it be interesting to see the performance on other diseases from the OpenNeuro datasets?
>
> **A:** Thank you for your constructive feedback. We acknowledge this is an interesting direction. To further evaluate the generalizability of our method beyond ADHD, we conduct additional experiments on the OpenNeuro-ds000030 dataset [1] for schizophrenia detection. The model achieves an average accuracy (ACC) of 81.39% across 10 random seeds, with a standard deviation of 4.91%; the average area under the curve (AUC) is 83.00%, with a standard deviation of 6.00%. The $p$-values for both metrics are well below 0.001 (ACC: $8.23 \times 10^{-9}$, AUC: $3.08 \times 10^{-8}$), indicating extremely high statistical significance and demonstrating that the model's performance significantly outperforms random guessing. The low relative standard deviation (approximately 7%–10%) also reflects the model's good stability under different initialization conditions. Overall, the model effectively captures brain imaging features associated with schizophrenia and holds potential clinical diagnostic value.
>
> [1] Bilder, R, Poldrack, R, Cannon, T, London, E, Freimer, N, Congdon, E, Karlsgodt, K, and Sabb, F (2020). UCLA Consortium for Neuropsychiatric Phenomics LA5c Study. OpenNeuro. [Dataset] doi: 10.18112/openneuro.ds000030.v1.0.0

---

> > ### Author Rebuttal · Reviewer_qPZM · 2026-04-04
> >
> > Thank you for including these new results I will keep my score but note my very low confidence.

---

> > > ### Author Response · Authors · 2026-04-07
> > >
> > > Thank you again for your careful reading of our work and for your continued support. We are glad that our response to your question was helpful, and we would like to take this opportunity to share a few additional points that we hope will give you a clearer and more complete picture of our work.
> > >
> > > **The Significance of Motivation in Neurodynamics.** Each component in SLT-BFGN is motivated by and validated against the empirical properties of the data rather than chosen arbitrarily. The short-term window design, for instance, is formally grounded in the first-order Markov property of brain functional connectivity, which we verify across 941 subjects from 7 independent ADHD-200 sites: lag-1 autocorrelation is $0.336 \pm 0.166$ while lag-2 decays to $-0.004 \pm 0.227$ (Markov ratio $= 4.76 \gg 1$, $p < 10^{-16}$, Appendix A). The contribution of each module is independently confirmed through comprehensive ablation studies (Table A, Reviewer zTtZ), where removing any individual component leads to a consistent and statistically significant performance drop.
> > >
> > > **Reproducibility of results.** All results reported in the paper are averaged over 10 independent runs with different random seeds under a fixed 60%/20%/20% stratified split, with mean and standard deviation explicitly reported alongside statistical significance tests (paired $t$-test, 95% CI) against a random classifier baseline, as detailed in Table A of our response to Reviewer zTtZ. The standard deviations are consistently low across all settings (e.g., $\pm1.93\%$ ACC and $\pm2.83\%$ AUC on ADHD-200 for the full model), indicating that the results are stable and not dependent on any particular initialization. We also commit to releasing the complete codebase, including preprocessing scripts, model implementations, and all hyperparameter configurations, upon acceptance, to ensure that the community can fully verify and build upon our results.
> > >
> > > **Validation across all subsets.** beyond the two primary datasets, we have validated our model across all 5 ADHD-200 acquisition sites and 13 OpenNeuro task subsets (Table B, Reviewer cmqV), all with statistically significant results, and on an entirely different disease (schizophrenia, OpenNeuro-ds000030), achieving ACC $= 81.39\% \pm 4.91\%$ and AUC $= 83.00\% \pm 6.00\%$ ($p < 0.001$). We hope this convergent evidence across multiple datasets, sites, seeds, and diseases provides a more solid basis for evaluating our work, and we would be very grateful if the reviewer could consider whether this additional context gives greater confidence in our results and conclusions.

---

### Decision · Program_Chairs · 2026-04-30

**Decision:**

Accept (regular)

**Comment:**

The paper proposes a dual-branch graph network for ADHD detection from fMRI data, integrating short-term brain dynamics with long-term structural information. Although all reviewers initially recommended rejection, post-rebuttal scores shifted upward to weak reject or weak accept. The integration of short- and long-term dynamics for ADHD detection is a reasonable direction, and the rebuttal meaningfully addressed several reviewer concerns.

The authors justify their architectural design by arguing that the signals exhibit a first-order Markov property. While this provides a theoretical basis for the dual-scale encoder, this critical motivation is currently developed only in the appendix rather than in the main text. The authors also provided ablation studies and temporal shuffling experiments to argue that temporal ordering is a key driver of performance. However, Reviewers JMdP and zTtZ remained unconvinced, characterizing the model as an "engineered solution" and questioning whether these justifications constitute a substantive technical contribution, although other reviewers found the Markov-based motivation and temporal experiments a reasonable justification.

Several concerns persist. Reviewers cmqV and JMdP continue to see limited novelty, arguing that the work largely integrates existing components, and they are uncertain whether the chosen fusion strategy is optimal. Reviewer JMdP also noted that the data partitioning is overly simple and lacks k-fold cross-validation. Reviewer zTtZ raised concerns about the use of BrainMVP, since it is pretrained on sMRI data. Additional issues were flagged regarding notational consistency throughout the manuscript. Reviewer qPZM was most positive, emphasizing clarity of presentation, while noting limited expertise in the specific subarea.

To strengthen the manuscript, the authors are encouraged to foreshadow the Markovian analysis and the neurodynamic motivation (the appendix) in the main text in order to properly justify the architecture. In addition, the authors are encouraged to incorporate the cross-validation results from the rebuttal, replacing Tables 1 and 2 and Figure 13 with central statistics (e.g., medians) and measures of dispersion (standard error or deviation), accompanied by tests of statistical significance to substantiate their claims.